# Multi-electron nitrobenzothiadiazole *sp*-conjugated-alkynyl covalent organic frameworks for ammonium-ion batteries

Yumin Chen[1], Da Zhang[1], Yang Qin[1], Chengmin Hu[2], Ling Miao[1], Yaokang Lv[3], Ziyang Song ●[1,4] ✉, Lihua Gan ●[1,5] ✉ & Mingxian Liu ●[1,5] ✉

Covalent organic frameworks containing periodic redox-active motifs and conjugation structures are booming as competitive negative electrodes for ammonium-ion batteries. Introducing substantial single-electron active motifs linked by dynamic imine bonds can increase their capacity; however, this design is constrained by suboptimal single-electron redox efficiency and insufficient linkage stability. Here we unlock a multiple two-electron-transfer nitrobenzothiadiazole covalent organic framework via integrating alkynyl benzenes and nitro-functionalized four-electron benzothiadiazoles. The high degree of π-electron *sp*-conjugation along alkynyl linkages and strong electron-drawing effect of nitrobenzothiadiazole motifs in nitrobenzothiadiazole covalent organic framework promise high $NH_4^+$ accessibility of multi-two-electron nitro/thiazole sites (95.2% utilization) with a lower activation energy (25.93 *vs.* 35.99 kJ mol$^{-1}$ of benzothiadiazole covalent organic framework).The fast octadeca-H-bonded $NH_4^+$ coordination in nitrobenzothiadiazole units liberates a high specific capacity of 317 mAh g$^{-1}$ for nitrobenzothiadiazole covalent organic framework negative electrode. The alkynyl-bridged π-conjugation network establishes structural anti-dissolution to enable a cycling durability of 70,000 cycles. Paired with high-voltage Prussian blue analogue positive electrode, the ammonium-ion full battery delivers a specific energy of 86.1 Wh kg$^{-1}$ (based on total active material mass) and a lifespan of 25,000 cycles. This work extends the design landscape of high-performance covalent organic frameworks for advanced ammonium-ion batteries.

Aqueous ammonium-ion batteries (AIBs) are gaining increasing attention as promising candidates for next-generation energy storage systems due to their intrinsic safety, low-cost, non-toxicity, and rapid reaction kinetics[1-6]. Compared with conventional metal-ion chemistries (e.g., Li$^+$, Na$^+$, and Zn$^{2+}$)[7-12], non-metallic $NH_4^+$ charge carrier possesses specific electrochemical properties of small hydrated ion radius (3.31 Å), low molar mass (18 g mol$^{-1}$), less corrosion and low possibility for hydrogen evolution reaction[13-17]. In addition, $NH_4^+$

[1]Shanghai Key Lab of Chemical Assessment and Sustainability, School of Chemical Science and Engineering, Tongji University, Shanghai, PR China. [2]Department of Chemistry, Laboratory of Advanced Materials, Shanghai Key Lab of Molecular Catalysis and Innovative Materials, Fudan University, Shanghai, PR China. [3]College of Chemical Engineering, Zhejiang University of Technology, Hangzhou, PR China. [4]State Key Laboratory of Pollution Control and Resource Reuse, College of Environmental Science and Engineering, Advanced Research Institute, Tongji University, Shanghai, PR China. [5]State Key Laboratory of Cardiovascular Diseases and Medical Innovation Center, Shanghai East Hospital, School of Medicine, Tongji University, Shanghai, PR China. ✉e-mail: songziyang@tongji.edu.cn; ganlh@tongji.edu.cn; liumx@tongji.edu.cn

delivers a preferential tetrahedral structure unlike metal ions, enabling stable H-bonding coordination chemistry with host materials[18–23]. Extensive efforts have been devoted to the development of electrode materials for efficient $NH_4^+$ storage in AIBs. Inorganic compounds (e.g., hexagonal $MoO_3$[24], 1 T/2H·$MoS_2$[25], and $Bi_2SeO_5$[26]) have been investigated as negative electrodes for AIBs. Nevertheless, their poor structural tunability and sluggish $NH_4^+$ kinetics within lattice structures pose significant challenges for the development of high-capacity and durable AIBs.

The alternative option was recently extended to π-conjugated aromatic organic negative electrode materials owing to their resource richness, broad-range structural and functional designability at the molecular level, allowing for systematic modulation of electrochemical energy storage performances[27–31]. In this regard, organic small molecules (e.g., 3,4,9,10-perylenetetracarboxylic diimide[32], 4,9,10-perylenetetra-carboxylic dianhydride[33]) have demonstrated promising $NH_4^+$ storage capability, benefiting from clearly defined high-mass-content ratio of redox sites that allow more electron transfer[34,35]. However, their high solubility and structural instability in aqueous electrolytes often result in rapid capacity fading and thus short life (<5000 cycles)[36–38]. To address this issue, researchers resorted to poorly soluble organic polymers for stable AIBs (up to 10,000 cycles). Unfortunately, the rotated polymeric chains and random stacking structures often lead to low exposure of redox-active motifs, limiting their capacity storage metrics (<200 mAh g$^{-1}$)[39,40].

Covalent organic frameworks (COFs), as crystalline porous polymers formed via periodically covalent linkage of π-conjugated building blocks, offer a unique structural platform to address the activity and stability limitations of organic molecules, uncontrolled polymers, and inevitable structure collapse of inorganic materials with ionic intercalation[41–47]. Their ordered architectures enable precise spatial organization of redox units, controlled pore environments, and enhanced charge transfer kinetics, making them ideal candidates for $NH_4^+$ storage[48–50]. To this end, there have been a few preliminary investigations on COFs negative electrodes for AIBs, such as quinone-pyrazine COFs and super-conjugated amine-linked anthraquinone COF materials[1,43,51]. These achievements broaden the design horizon of COFs to boost the capacity of AIBs, but generally requires the introduction of substantial single-electron active motifs (e.g., C=O, C=N) based on dynamic amine bond coupling. Unfortunately, this strategy has almost reached the capacity saturation point of COFs (<250 mAh g$^{-1}$) under limited redox efficiency, at the same time it brings structural instability caused by twisted −NH− linkages, resulting in unsatisfactory cycling life (<10,000 cycles). To break the efficiency limitation of single-electron site reactions, we envision creating multiple two-electron redox-active motifs into COFs with robust structural linkages via strategically structural engineering, thereby unlocking more stable double-electron redox transfer to reform AIBs with better $NH_4^+$-storage capacity and life, but this has not yet been achieved.

In this work, we report a two-electron-nitro modulated nitro-benzothiadiazole COF (nitro-BTH-COF) negative electrode material by fusing alkynyl (−C≡C−)-bridged benzenes and four-electron-accepting nitrobenzothiadiazole units. nitro-BTH-COF exhibits extended π-electron conjugation through alkynyl linkages and strong electron-withdrawing nitrobenzothiadiazole motifs, enabling high $NH_4^+$-accessible multi-two-electron nitro/thiazole sites (95.2% utilization) with reduced activation energy (25.93 vs. 35.99 kJ mol$^{-1}$ of BTH-COF). This favorable architecture activates a rapid and stable octadeca-H-bonded $NH_4^+$ storage mechanism via H-bonding coordination electrochemistry, yielding a high specific capacity for organic negative electrodes in AIBs. Furthermore, the robust alkynyl-bridged π-conjugation network imparts structural integrity of nitro-BTH-COF negative electrode and anti-dissolution in aqueous electrolyte to achieve stable cycling performance. When integrated with Prussian blue (NiFeHCF) positive electrode, the assembled nitro-BTH-COF‖NiFeHCF

full cell demonstrates practical specific energy and stable cycling durability. This work pioneers a promising design strategy of multi-electron redox and stable COFs for advanced energy storage.

## Results and discussion

### Materials preparation and characterization

nitro-BTH-COF was designed by Sonogashira coupling of 1,3,5-triethynylbenzene (TEB) and 4,7-dibromo-5,6-dinitro-2,1,3-benzothiadiazole (nitro-BTH) molecules (Fig. 1a), while BTH-COF was synthesized by coupling TEB and 4,7-dibromo-2,1,3-benzothiadiazole (BTH) monomers[52,53]. Alkynyl (−C≡C−) modules of TEB units as the linkers in BTH-COF offer stable sp-conjugated skeleton, which is expected to improve structural robustness and anti-dissolution in aqueous electrolytes, while thiazole motifs of BTH units serves as two-electron active sites to couple with $NH_4^+$ charge carries (Supplementary Fig. 1). In contrast, the introduction of strong electron-withdrawing nitro groups into nitro-BTH-COF brings extra two-electron redox-active motifs, while inheriting stable sp-conjugated alkynyl (−C≡C−) modules. Compared with C=C bond-forming strategies that easily induce the spatial structure distortion in COFs, sp-conjugated alkyne (−C≡C−) linkages via Sonogashira coupling enables the formation of rigid and robust skeletons to minimize structural deformation of nitro-BTH-COF, which is expected to establish structural anti-dissolution in aqueous electrolytes for durable electrochemical reactions.

Electrostatic potential (ESP) simulations[54], suggest obviously more negative charge distributions around two-electron-transfer nitro/thiazole motifs in nitro-BTH (5.13 Debye dipole moment) in comparison to BTH (Fig. 1b). nitro-BTH-COF shows a high degree of extended sp-conjugation along alkynyl linkages, as demonstrated by the π-electron localization function map (ELF-π, Supplementary Fig. 2). These properties endow nitro-BTH-COF with strong ammonia-philic ability and π-electron delocalization effect. Generally, a highly conjugated structure results in a reduction in the energy levels of the lowest unoccupied molecular orbital (LUMO) and the highest occupied molecular orbital (HOMO)[55]. The LUMO-HOMO gap (ΔE) of nitro-BTN-COF (1.48 eV from B3LYP-D3/def2-SVP) is lower than BTN-COF (1.54 eV from B3LYP-D3/TZVP, Supplementary Fig. 3), promising high electron transfer efficiency. Thus, nitro-BTN-COF integrates multi-two-electron nitro/thiazole active sites, highly π-electron conjugation structure, and low energy barriers.

Experimental powder X-ray diffraction (PXRD) patterns of BTH-COF and nitro-BTN-COF are presented in Fig. 1c, d, respectively. The experimental PXRD patterns of both BTH-COF and nitro-BTH-COF match the simulated AA-stacking models (rather than AB-stacking models) in terms of peak positions and relative intensities (Fig. 1c, d and Supplementary Fig. 4). Furthermore, the experimental pore sizes of BTH-COF (1.90 nm) and nitro-BTH-COF (1.72 nm) analyzed by nitrogen adsorption/desorption are consistent with those calculated by the AA-stacking model (1.81 and 1.67 nm, Supplementary Fig. 5). Scanning electron microscope (SEM) and high-resolution transmission electron microscopy (HRTEM) images reveal the dendritic structures of nitro-BTH-COF (Fig. 1e), together with highly ordered crystalline frameworks, uniform element distributions, porous structures and structural stability (Supplementary Figs. 6–8).

Fourier transform infrared (FT-IR) spectra were performed to identify the functional structures of nitro-BTN-COF (Fig. 1f). The characteristic peaks at 2155, 1654, 1571 and 1269/1473 cm$^{-1}$ can be attributed to C≡C bonds, C=N groups, C=C stretching vibrations, and nitro motifs, respectively[41,52,56–58], confirming its successful synthesis from TEB and nitro-BTH monomers. Furthermore, solid-state $^{13}$C nuclear magnetic resonance (NMR) analysis of nitro-BTN-COF shows chemical shifts at 157.2, 140.8, 130.0, and 109.6 ppm (Fig. 1g and Supplementary Fig. 9a), which belong to the carbon atom from nitrobenzothiadiazole units and benzene rings[59]. The signals at 93.2 and 80.2 ppm can be ascribed to the alkynyl group connected to

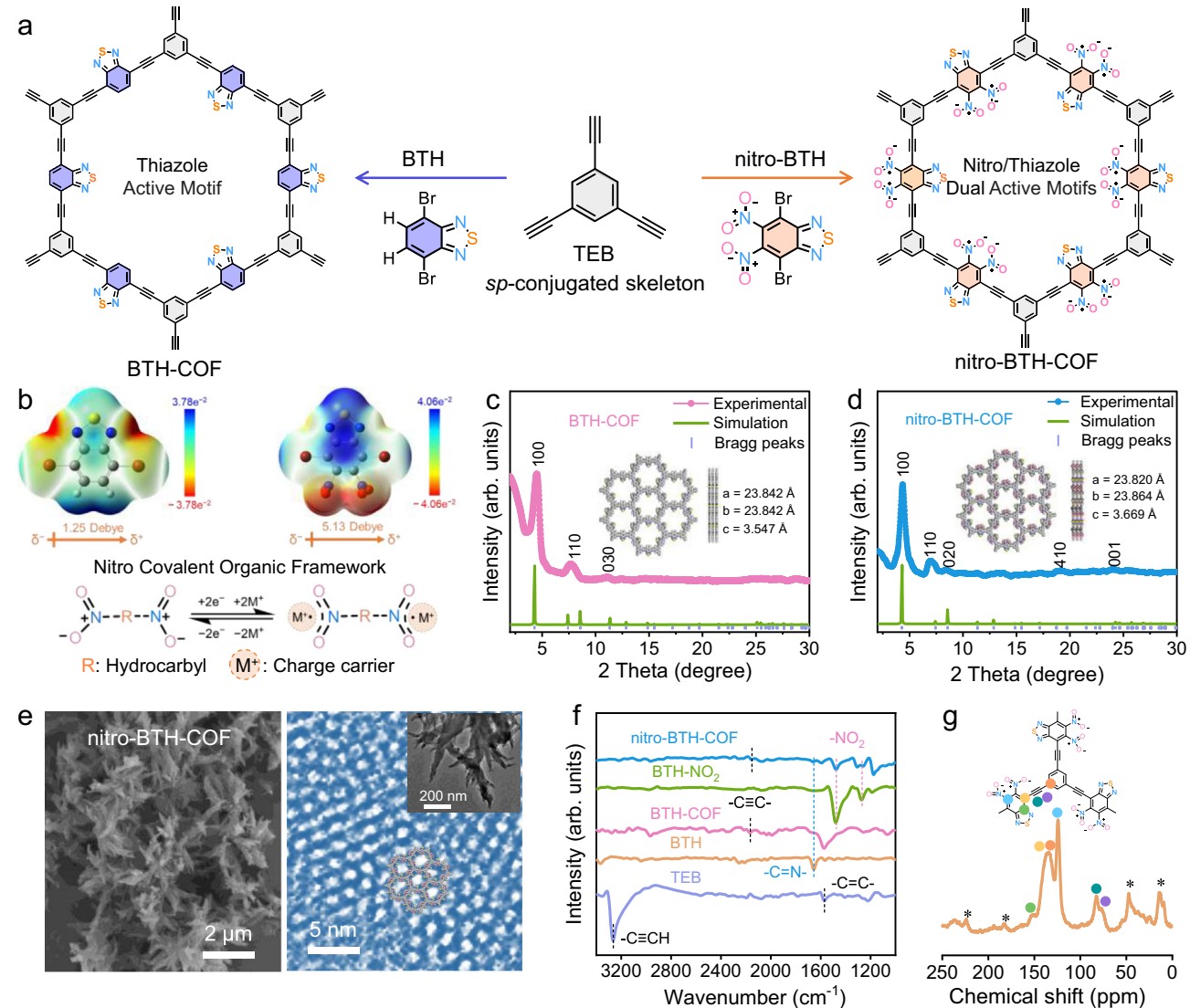

**Fig. 1 | Material synthesis and characterization. a** Schematic synthesis of BTH-COF and nitro-BTH-COF. **b** ESP maps, dipole moments, and two-electron transfer mechanisms of nitro sites. Colors of elements: H, white; C, gray; O, red; N, blue; S, yellow; Br, brownish-red. Experimental and simulated PXRD patterns of **c** BTH-COF and **d** nitro-BTH-COF. **e** SEM and HRTEM images of nitro-BTH-COF. **f** FT-IR spectra. **g** Solid-state $^{13}$C NMR spectrum of nitro-BTH-COF.

benzene rings and benzothiadiazole units, respectively. Compared to BTH-COF, the high-resolution N 1s X-ray photoelectron spectrum (XPS) of nitro-BTH-COF exhibits an additional peak assigned to $NO_2$ (Supplementary Fig. 9b), confirming the successful introduction of nitro groups. All these results indicate the successful fabrication of *sp*-conjugated nitro-BTN-COF with alkynyl linkages.

## Electrochemical performance and kinetics analysis

To investigate the electrochemical performances of nitro-BTH-COF as the working electrode, a three-electrode Swagelok cell was assembled by using Ag/AgCl as the reference electrode, activated carbon as the counter electrode and aqueous 2 M $(NH_4)_2SO_4$ solution as the electrolyte (Supplementary Fig. 10a). In cyclic voltammetry (CV) profiles, three pairs of negative redox peaks at −0.71/− 0.57, −0.38/− 0.34, and −0.22/− 0.10 V are observed for nitro-BTH-COF negative electrode, while only a pair of redox peaks at −0.72/− 0.68 V for BTH-COF (Fig. 2a and Supplementary Fig. 10b−c). Obviously, the incorporation of nitro groups into nitro-BTH-COF introduces additional redox-active sites and enhances $NH_4^+$ reactivity. Moreover, nitro-BTH-COF negative electrode delivers high capacities of 317 mAh g$^{-1}$ at 0.2 A g$^{-1}$ and

135 mAh g$^{-1}$ at 50 A g$^{-1}$ (Fig. 2b, c), significantly outperforming BTH-COF negative electrode (117/42 mAh g$^{-1}$ at 0.2/50 A g$^{-1}$, Supplementary Fig. 10d). When the mass loading of nitro-BTH-COF negative electrode increases from 2.1 to 10.2 mg cm$^{-2}$, it still shows a high capacity of 246 mAh g$^{-1}$ (Supplementary Fig. 11), demonstrating its excellent commercial application potential.

nitro-BTH-COF negative electrode at 0.2 A g$^{-1}$ achieves better capacity retention over extended cycles than that of BTH-COF (Supplementary Fig. 12a−b), due to the attenuated activity and reversibility of thiazole active sites[2]. In addition, nitro-BTH-COF negative electrode at 6 A g$^{-1}$ delivers a high-capacity retention of 94.59% over 20,000 cycles (Supplementary Fig. 12d), showing its desirable electrochemical stability. Even at a high specific current of 20 A g$^{-1}$, nitro-BTH-COF negative electrode demonstrates long-term stability with 93.70% capacity retention after 70,000 cycles (Fig. 2d), a performance that is compared with other reported organic/inorganic electrode materials in AIBs (Fig. 2e and Supplementary Table 1)[1,3,21,43,51,60−64]. Overall, thanks to the high degree of π-electron *sp*-conjugation along alkynyl linkages and strong electron-drawing effect of nitrobenzothiadiazole motifs, nitro-BTH-COF negative electrode liberates better comprehensive

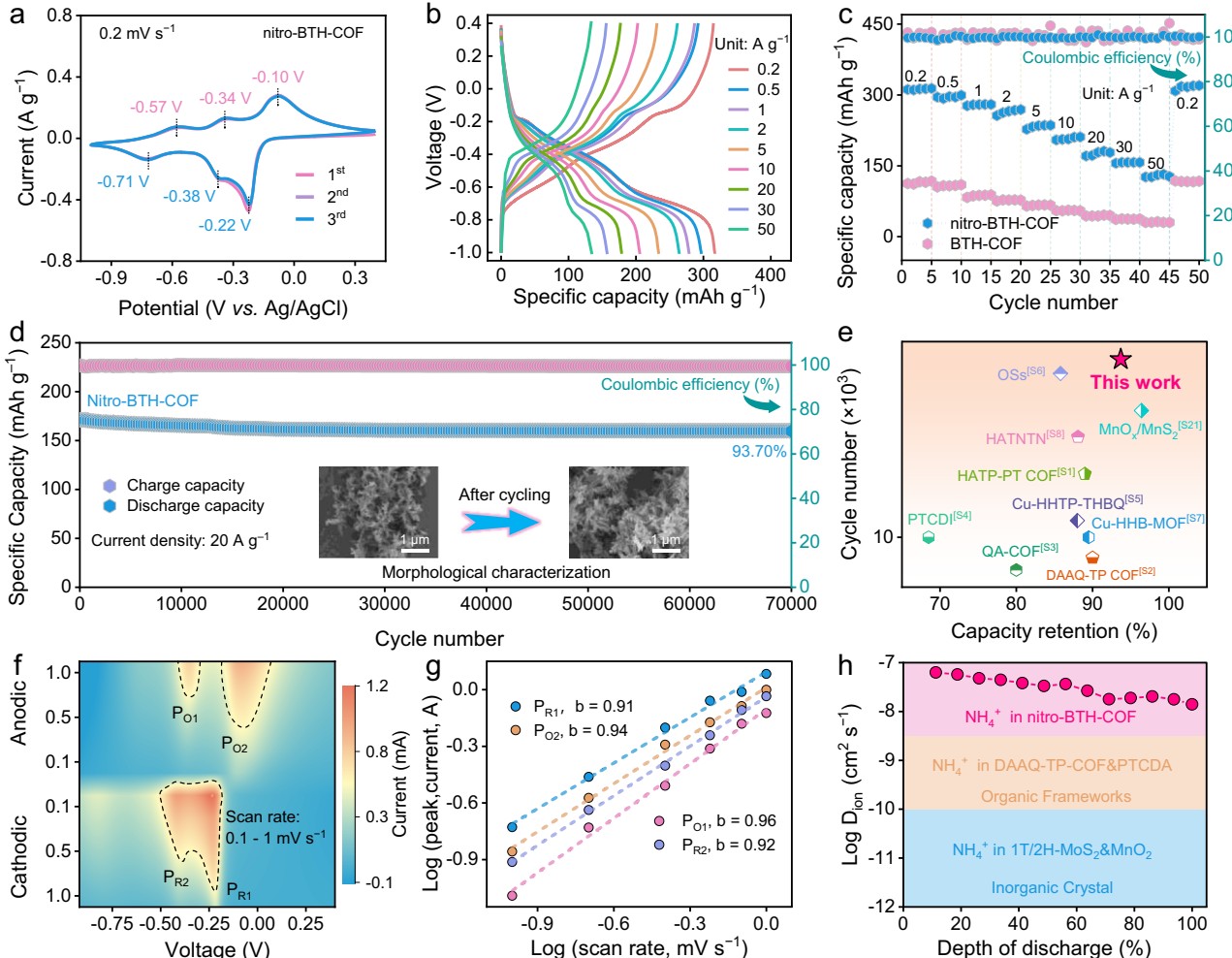

**Fig. 2 | Electrochemical performances of nitro-BTH-COF and BTH-COF electrodes.** All electrochemical tests were conducted at 25 ± 0.5 °C under ambient pressure. **a** CV curves. **b** GCD profiles. **c** Rate capacities. **d** Cycling performance (insets: SEM images of nitro-BTH-COF electrode in the fully discharged state before and after 70,000 cycles). **e** Lifetime comparison of nitro-BTH-COF and reported organic materials (The source of the literature data shown in this figure can be found in Supplementary Information, Table 1). Charge storage kinetics of nitro-BTH-COF negative electrode. **f** Contour plots of CV patterns at different scan rates. **g** Calculated b values. **h** Comparison of evaluated NH$_4^+$ diffusion coefficient during discharging and reported electrodes.

performance in terms of specific energy, capacity, rate performance, and cycle stability than that of BTH-COF (Supplementary Fig. 12d), holding desirable electrochemical potential for propelling AIBs. Post-cycling SEM, XRD and FT-IR characterizations confirm the structural and functional integrity of nitro-BTH-COF (insert of Fig. 2d and Supplementary Fig. 13). The all-round electrochemical performances make nitro-BTH-COF a promising negative electrode for AIBs.

The electrochemical redox kinetics of NH$_4^+$ storage in nitro-BTH-COF negative electrode was systematically investigated through CV patterns at varying scan rates (0.1 – 1 mV s$^{-1}$). Compared with BTH-COF, two extra distinct reduction peaks at −0.30 V (P$_{R1}$) and −0.48 V (P$_{R2}$) in CV profiles correspond to the sequential NH$_4^+$ coordination with nitro redox centers (Fig. 2f), demonstrating good electrochemical reversibility. Quantitative kinetics analysis based on the power-law relationship ($i = kv^b$) yields remarkably high b-values of 0.91 − 0.96, confirming surface-dominated charge storage processes (Fig. 2g). The capacitive contribution accounts for 71.23% of total charge storage at 0.1 mV s$^{-1}$ and increases to 93.31% at 1 mV s$^{-1}$, highlighting its fast redox kinetics (Supplementary Fig. 14a−b). Galvanostatic intermittent titration technique (GITT) analysis further confirms the high NH$_4^+$ diffusion coefficient of 10$^{-8}$ ∼ 10$^{-7}$ cm$^2$ s$^{-1}$ for nitro-BTH-COF negative electrode (Fig. 2h and Supplementary Fig. 14c), surpassing previously reported values in

organic/inorganic hosts (∼10$^{-12}$ − 10$^{-8}$)[25,51,61,65]. Overall, nitro-BTH-COF negative electrode exhibits favorable electrochemical performances in terms of specific capacity, rate capability and cycling stability, which stem from the synergy of multiple redox-active nitro/thiazole groups and alkynyl-linked $sp$-conjugated robust framework that facilitates high-kinetics and stable NH$_4^+$ storage.

## Structural evolution and mechanism understanding

To elucidate the NH$_4^+$ storage mechanism in nitro-BTH-COF negative electrode, comprehensive spectroscopic characterizations were conducted to monitor its structural evolution at various (dis)charge potentials (Fig. 3a). In FT-IR spectra (Fig. 3b), the vibration signal of nitro motifs (−NO$_2$) at 1293/1471 cm$^{-1}$ gradually decreases during discharge accompanied by the emergence of a new peak at 1206 cm$^{-1}$, corresponding to NH$_4^+$-coordinated nitro species of nitro-BTH-COF negative electrode ([O−N·−O]···NH$_4^+$)[41]. Simultaneously, the observed reversible intensity variations at 1651 cm$^{-1}$ demonstrate NH$_4^+$ interaction with C=N groups of thiazole units, while the appearance and subsequent disappearance of a H-bond stretching mode (NO$_2$/C=N···NH$_4^+$) at 2928 cm$^{-1}$ during discharge/charge provide direct evidence for dynamic H-bonding electrochemistry between NH$_4^+$ and nitro-BTH-COF framework[66]. The persistent signal at 1586 cm$^{-1}$

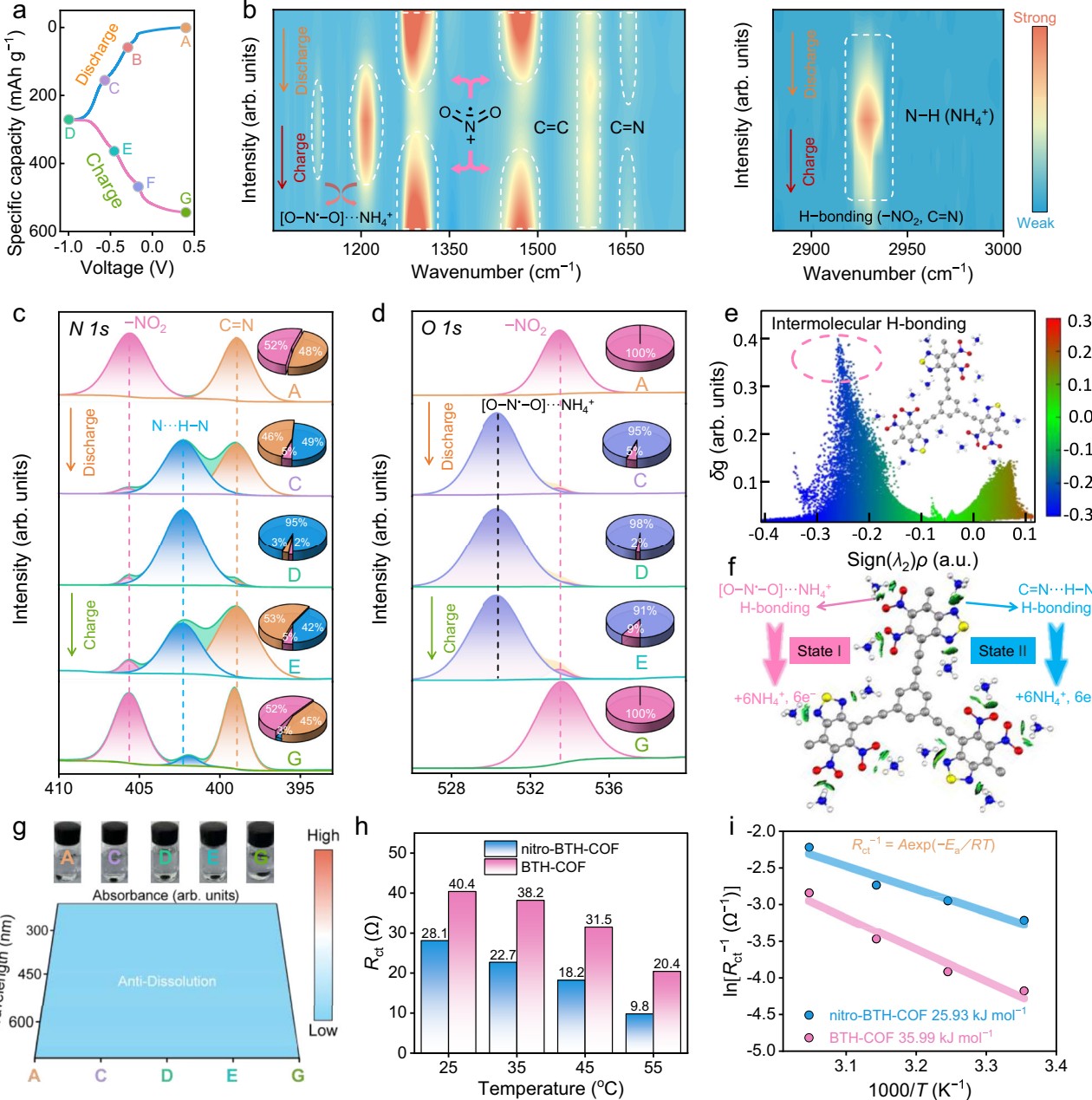

**Fig. 3 | Charge-storage behavior of nitro-BTH-COF electrode. a** A GCD profile with specific (dis)charge states. **b** Overview of FT-IR spectra. Ex-situ XPS spectra of **c** *N 1 s* and **d** *O 1 s*. For ex situ spectrum analysis, cells were cycled at $1 A g^{-1}$ for 3 cycles at $25 \pm 0.5\,°C$ under ambient pressure, and terminated at designated states of charge/discharge before disassembly. **e** Plots of IGM versus $sign(\lambda_2)\rho$ and corresponding gradient isosurfaces. **f** H-bonding $NH_4^+$ coordination mechanism. Colors of elements: H, white; C, gray; O, red; N, blue; S, yellow. **g** UV/Vis spectra and photos of nitro-BTH-COF at different (dis)charge states soaked in $2 M (NH_4)_2SO_4$ aqueous electrolyte. **h** Evaluated $R_{ct}$ values. **i** Arrhenius plots of $\ln(R_{ct}^{-1})$ against $1000/T$ and calculated $E_a$ values.

corresponding to C≡C bonds confirm the structural integrity of the *sp*-conjugated scaffold of nitro-BTH-COF throughout $NH_4^+$ (de)coordination. These spectroscopic signatures demonstrate reversible H-bonding $NH_4^+$ redox storage reactions of $NO_2$/C=N groups in *sp*-conjugated robust matrix.

XPS spectrum analysis of nitro-BTH-COF negative electrode was conducted to investigate the H-bonding chemistry between $NH_4^+$ ions and $NO_2$/C=N sites in nitro-BTH-COF. The evolution of N 1 s spectra clearly demonstrates a stepwise $NH_4^+$ coordination mechanism. At the initial state A, two characteristic peaks are observed at 405.6 eV ($NO_2$) and 398.9 eV (C=N) (Fig. 3c)[67]. During the first discharge stage (state A → B → C), the peak intensity of $NO_2$ progressively decreases while the

C=N signal remains essentially unchanged, indicating preferential $NH_4^+$ bonding with $NO_2$ groups. As the discharge process continues (state C → D), the C=N peak shows significant attenuation while a new peak emerges at 402.3 eV, corresponding to the H-bonding formation of H−N···N (C=N). This sequential coordination behavior is fully reversible during the subsequent charging process (state D → G), with all signals returning to their initial states (Supplementary Fig. 15). O 1 s spectra further reveal the formation of [O−N·−O]···$NH_4^+$ species during discharge (Fig. 3d), confirming the H-bonding $NH_4^+$ coordination mechanism.

Besides, the independent gradient model (IGM) as a function of the density across nitro-BTH-COF was plotted to depict their non-

covalent interaction (Fig. 3e). An obvious blue spike within the range of −0.3 to −0.2 a.u. of the sign $(\lambda_2)\rho$ suggest the strong H-bonding interaction between nitro-BTH-COF (H-bond acceptor) and $NH_4^+$ (H-bond donor). Overall, two conclusions can be drawn: (i) $NO_2$ and C=N groups are identified as the dual redox-active sites to afford the whole H-bonding electrochemistry process of nitro-BTH-COF negative electrode; (ii) the charge storage mechanism during discharging involves a successive two-stage $NH_4^+$ coordination process with two-stage $NO_2$ first (state A → B → C) followed by C=N motifs (state C → D) (Fig. 3f and Supplementary Fig. 16).

UV/vis spectroscopy was employed to investigate the dissolution behavior of nitro-BTH-COF negative electrode at various (dis)charge states. The absence of distinct absorption peaks in the spectra (Fig. 3g), coupled with colorless $(NH_4)_2SO_4/H_2O$ electrolytes, confirms the structural insolubility of nitro-BTH-COF. Electrochemical impedance spectroscopy (EIS) measurements at different temperatures (Supplementary Fig. 17) enable the determination of the activation energy ($E_a$) for interfacial charge transfer through Arrhenius equation[68,69]. The obtained $E_a$ values reflect the energy barrier for the interfacial coordination between redox-active groups of organics and $NH_4^+$ charge carriers. Generally, the charge transfer resistances ($R_{ct}$, 9.8 − 28.1 Ω) of nitro-BTH-COF negative electrode are lower than that of BTH-COF (20.4 − 40.4 Ω, Fig. 3h and Supplementary Table 2), implying faster reaction kinetics. Notably, nitro-BTH-COF exhibits a lower $E_a$ value of 25.93 kJ mol⁻¹ (Fig. 3i) compared to BTH-COF (35.99 kJ mol⁻¹). Such a reduced energy barrier does a favor to high-kinetics interfacial $NH_4^+$ coordination with $NO_2$/C=N redox sites of nitro-BTH-COF, enabling fast charge storage processes.

## Theoretical calculation and dynamic simulation

Density functional theory (DFT) simulations were systematically performed to elucidate the fundamental charge storage mechanism of nitro-BTH-COF during $NH_4^+$ uptake/removal processes. Structural optimization analyses revealed that $NH_4^+$ ions preferentially form thermodynamically stable coordination structures with two nitro oxygen motifs and a C=N site in the molecular plane (Supplementary Fig. 18), enabling octadeca-H-bonding interactions per nitrobenzothiadiazole unit and thus twelve-electron redox reactions that correspond to a theoretical capacity of 332.9 mAh g⁻¹. A comprehensive thermodynamic investigation was conducted to determine the $NH_4^+$ coordination pathways, where nitro groups exhibit significantly stronger ammonia-philic character compared to C=N sites (Supplementary Fig. 18). Based on the principle of minimum energy, we established a sequential two-stage 12-electron transfer mechanism for $NH_4^+$ storage in nitro-BTH-COF (Fig. 4a). The Gibbs free energy ($\Delta G$) analysis reveals distinct coordination behaviors in nitro-BTH-COF (Fig. 4b): (i) For up-front $NH_4^+$ uptake (Initial state → state I, nitro-BTH-COF-6NH$_4^+$), the energy map ($\Delta G_{1-1} < \Delta G_{1-2} < \Delta G_1$) clearly indicates a two-sequential coordination process involving six $NH_4^+$ ions at nitro sites. (ii) During subsequent $NH_4^+$ incorporation (state I → state II, nitro-BTH-COF-12NH$_4^+$), the energy trend ($\Delta G_2 < \Delta G_{2-1} < \Delta G_{2-2}$) suggests a direct one-step coordination process for the remaining six $NH_4^+$ ions at C=N sites. The proposed two-stage twelve-electron $NH_4^+$ coordination mechanism in nitro-BTH-COF agrees with three redox voltages of CV and GCD profiles (Fig. 2a–b), demonstrating fast and robust charge storage.

Differential charge density isosurfaces clearly visualize the binding characteristics between $NH_4^+$ ions and nitro-BTH-COF (Fig. 4c). Significant charge accumulation/depletion around $NO_2$/C=N sites confirm strong $NH_4^+$ interactions, with obviously Bader charge transfers (State I: 0.63 e; State II: 0.92 e). The localized orbital locator-π (LOL-π) map demonstrates that $NH_4^+$ coordination induces local polarization, the $sp^2$-$sp$ hybridized framework maintains effective π-conjugation throughout the storage process (Supplementary Fig. 19). Both nitro-BTH-COF-6NH$_4^+$ (state I) and nitro-BTH-COF-12NH$_4^+$ (state

II) show efficient $NH_4^+$-mediated electron delocalization through the whole skeleton (Fig. 4d–e). Frontier molecular orbital analysis reveals charge-storage intermediates possess ideal electronic properties for $NH_4^+$ storage, exhibiting both strong electron affinity (progressively increasing LUMO levels) and favorable charge transfer capability (narrow HOMO-LUMO gaps of 0.58, 0.66, 0.71 eV from B3LYP-D3/TZVP, Fig. 4f). $NH_4^+$ binding in nitro-BTH-COF via multiple H-bonding interactions (N−H···O and N−H···N) introduces increased steric congestion that restricts structural relaxation[41,70,71]. This drives an upward shift in both HOMO and LUMO energy levels and results in a modest widening of the energy gap (from 0.58 to 0.71 eV). These results indicate the favorable electronic structures including multi-two-electron nitro/thiazole active sites, highly π-electron conjugation structure, and low energy barriers, which enables high-kinetics and stable stepwise $NH_4^+$ coordination to achieve high-performance $NH_4^+$ storage.

Nuclear independent chemical shift (NICS) and Harmonic oscillator model of aromaticity (HOMA) analysis, were applied to quantitatively reflect the molecular aromaticity[69,72]. Upon the uptake of $NH_4^+$ ions on nitro-BTH-COF, HOMA values close to 1, coupled with negative NICS values, reflect a pronounced aromatic character and extensive π-electron delocalization across each intermediate state, suggesting great skeleton aromaticity and structural robustness (Fig. 4g). Molecular dynamics (MD) simulations were performed on nitro-BTH-COF to better understand the H-bonding formation upon $NH_4^+$ coordination (Fig. 4h and Supplementary Fig. 20). The nitro/C=N motifs of nitro-BTH-COF can offer multiple ammonia-philic active sites to induce high-density H-bonding network (average: 212.26, Fig. 4i) compared to BTH-COF (107.04). These results confirm that $NH_4^+$-coordination triggers charge flow to allow for high availability of dual $NO_2$/C=N motifs and multielectron redox reactions, bringing boosted electrochemical activity and durability.

## Electrochemical performance of the full battery

The $NH_4^+$ storage performance of nitro-BTH-COF negative electrode was further investigated in a nitro-BTH-COF‖NiFeHCF full-cell configuration with NiFeHCF positive electrode (Supplementary Fig. 21) and 2.0 M $(NH_4)_2SO_4$ electrolyte (Fig. 5a). As shown in Fig. 5b, nitro-BTH-COF negative electrode demonstrates a stable operating potential window of −1.0 - 0.4 V (vs. Ag/AgCl), while NiFeHCF positive electrode exhibits a complementary potential range of 0.4 - 1.20 V. Rocking-chair nitro-BTH-COF‖NiFeHCF battery displays three pairs of well-defined redox peaks with minimal polarization (Fig. 5c), indicating highly reversible multielectron $NH_4^+$ storage behaviors and fast diffusion kinetics in both electrode materials. The nitro-BTH-COF‖NiFeHCF battery delivers a high capacity of 310 mAh g⁻¹ at 0.2 A g⁻¹ (based on mass loading of negative electrode, Fig. 5d). Even at 10 A g⁻¹, the capacity can still reach 206 mAh g⁻¹ (Fig. 5e).

Based on the total mass loadings of active materials in nitro-BTH-COF negative electrode (2.2 mg cm⁻²) and NiFeHCF positive electrode (5.2 mg cm⁻²), the high capacity (78 mAh g⁻¹) and high average output voltage (1.1 V) bring a state-of-the-art battery-level specific energy of 86.1 Wh kg⁻¹ among all reported $NH_4^+$ full batteries (Fig. 5f and Supplementary Table 3)[43,51,61,73,74]. Furthermore, nitro-BTH-COF‖NiFeHCF battery achieves an extraordinary cycling lifespan of 25,000 cycles with 88.91% capacity retention at 10 A g⁻¹ (Fig. 5g), which is the highest value among reported $NH_4^+$ full batteries (Fig. 5h). XRD patterns of inorganic NiFeHCF positive electrode shows the gradually weaken crystal peaks of (420), (422), and (620) planes (Supplementary Fig. 21a), indicating its destroyed crystalline structure during the long-term $NH_4^+$-insertion/extraction cycling process. Given the structure stability of nitro-BTH-COF (Supplementary Fig. 21b), the 11.09% capacity loss of nitro-BTH-COF‖NiFeHCF battery can be attributed to the crystal structural degradation of NiFeHCF positive electrode after 25,000 cycles. Besides, three integrated nitro-BTH-COF‖NiFeHCF

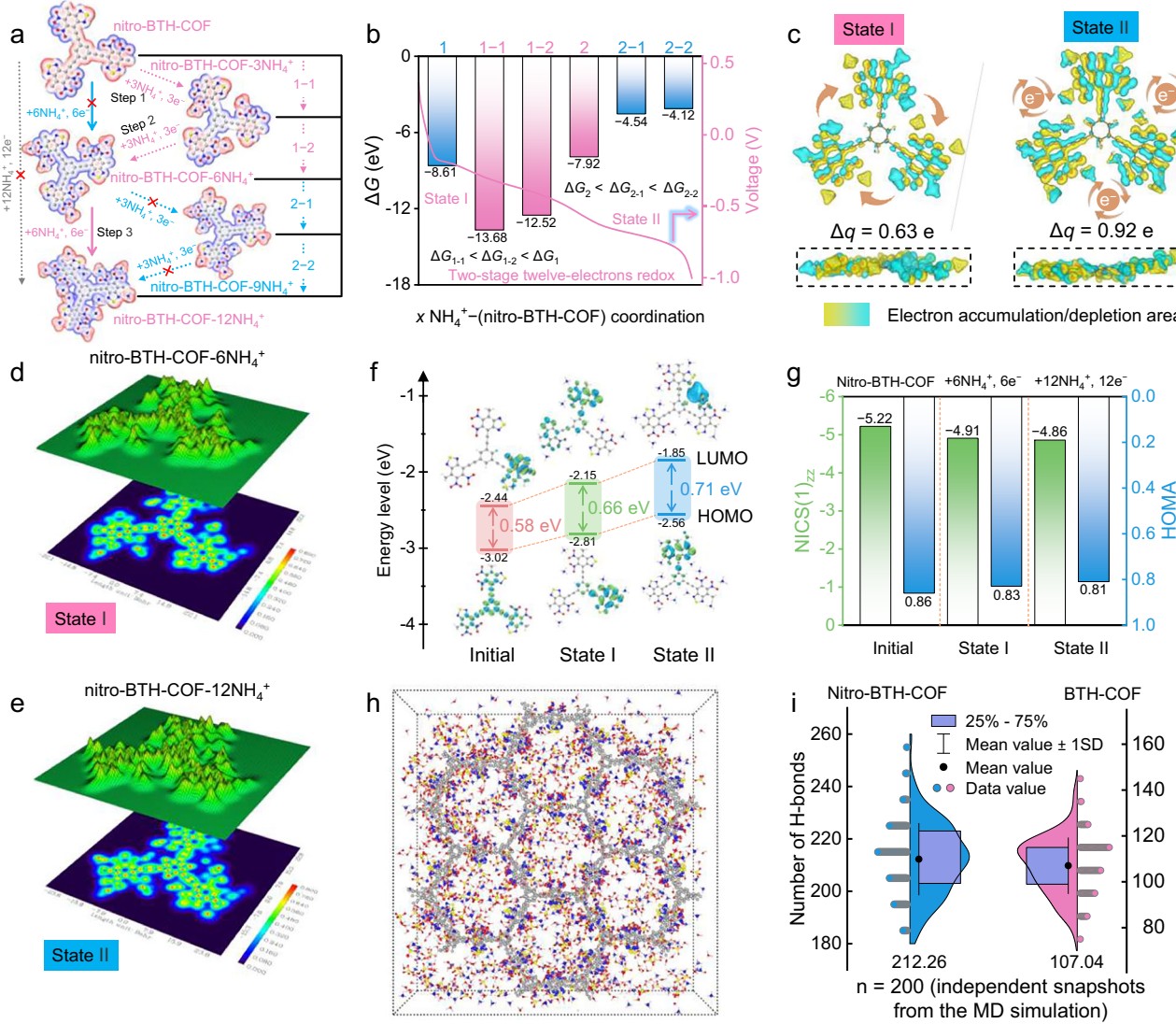

**Fig. 4 | Theoretical simulations and energy storage mechanism of nitro-BTH-COF. a** Structural evolution and MEP simulation of nitro-BTH-COF under various $NH_4^+$ coordination routes. **b** Calculated $\Delta G$ values of possible reaction paths. **c** Charge density difference isosurfaces of nitro-BTH-COF coordinated with six $NH_4^+$ ions (state I) and twelve $NH_4^+$ ions (state II), respectively. LOL-π and corresponding topographic maps of **d** state I and **e** state II. **f** Frontier molecular orbital diagrams and energy levels of nitro-BTH-COF at different states. **g** NICS and HOMA values. **h** Molecular dynamics simulation snapshots of $NH_4^+$-coupled nitro-BTH-COF and **i** statistics of the corresponding number of H-bonds (Data are presented as mean values ± standard deviation derived from $n = 200$ independent snapshots extracted from the final 200 ps of the MD simulation). Colors of elements: H, white; C, gray; O, red; N, blue; S, yellow.

batteries (3.7 V) can power a light emitting diode toy (inset of Fig. 5g), demonstrating practical viability.

In conclusion, multiple two-electron nitro-BTH-COF is designed by fusing alkynyl benzenes and nitrobenzothiadiazole units. nitro-BTH-COF delivers highly π-electron *sp*-conjugation along alkynyl linkages and strong electron-drawing nitrobenzothiadiazole motifs. This feature promises high $NH_4^+$ utilization (95.2%) of multi-two-electron nitro/thiazole sites with a lower activation energy (25.93 *vs.* 35.99 kJ mol⁻¹ of BTH-COF). Experiment and theorical studies reveal the fast and stable octadeca-H-bonded $NH_4^+$ coordination mechanism of nitro-BTH-COF negative electrode. As a consequence, nitro-BTH-COF negative electrode offers competitive capacity among reported COFs in AIBs. Furthermore, the alkynyl-bridged π-conjugation network of nitro-BTH-COF negative electrode ensures the structural insolubility in aqueous electrolyte, affording stable cycling life. By pairing nitro-BTH-COF negative electrode with high-voltage Prussian blue positive electrode, the constructed nitro-BTH-COF‖NiFeHCF full battery demonstrates favorable specific energy and cycling life. This work gives in-depth insights into the design of multi-electron-redox and stable COFs for advanced AIBs.

## Methods

### Materials

All chemicals were commercially available and used without further purification. 1,3,5-triethynylbenzene (TEB, 99%), 4,7-dibromo-2,1,3-benzothiadiazole (BTH, 99%), 4,7-dibromo-5,6-dinitro-2,1,3-benzothiadiazole (nitro-BTH, 99%), copper (I) iodate (CuI, 99.9%) tetrakis (triphenylphosphine) palladium (Pd(PPh₃)₄, 99.9%) N,N-dimethylformamide (DMF, 99.8%), N-methyl-2-pyrrolidone (NMP, 99.9%), triethylamine (Et₃N, 99.5%) and methanol (MeOH, 99.9%) were purchased from Adamas-Beta. Polyvinylidene difluoride (PVDF, Mw = 1200000 Da, 99.5%), Super P (particle size: 40 – 50 nm, 99.5%), separator (Whatman CF/D, thickness: 675 μm, lateral dimension: 90 mm, porosity: 90 ± 2%, pore size: 2.7 μm), coin cell components

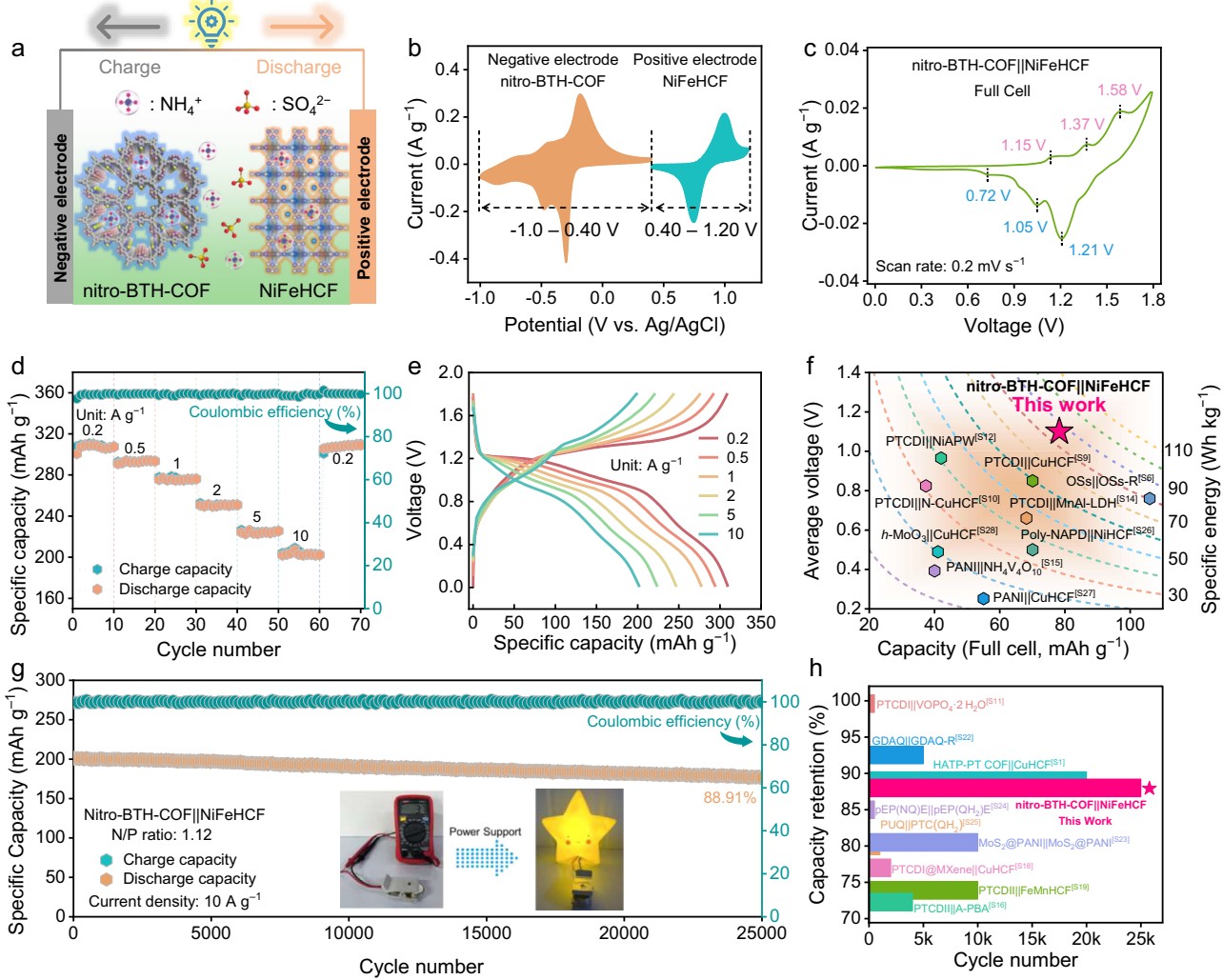

**Fig. 5 | Electrochemical performance of nitro-BTH-COF‖NiFeHCF full battery.** All electrochemical tests were conducted at 25 ± 0.5 °C under ambient pressure. **a** Configuration and operation mechanism. **b** CV profiles of nitro-BTH-COF negative electrode and NiFeHCF positive electrode. **c** CV curves, **d** rate performances, and **e** GCD curves of the full cell. **f** Voltage-capacity contour plots, **g** cycling stability, and **h** lifespan comparison of reported AIBs. The source of the literature data shown in this **f** and **h** can be found in Supplementary Information, Table 3.

(CR2032, positive electrode case: 18 × 2.91 mm, negative electrode case: 16 × 2.77 mm, spacer: 15.8 × 1 mm, spring: 15.6 × 1.1 × 0.25 mm) were purchased from the Guangdong Canrd New Energy Technology Co., Ltd. Carbon felt (thickness: 0.05 mm) was purchased from Wuhu Eryi Material Technology Co., Ltd. Dichloromethane ($CH_2Cl_2$, 99.8%), potassium chloride (KCl, 99.9%), ferrous sulfate heptahydrate ($FeSO_4 \cdot 7H_2O$, >99.0%) and nickel(II) sulfate heptahydrate ($NiSO_4 \cdot 7H_2O$, >99.0%) were purchased from Sigma-Aldrich.

## Synthesis of BTH-COF and nitro-BTH-COF
In a typical procedure, 0.4 mmol of TEB and 0.6 mmol of either BTH or nitro-BTH were combined with CuI (2.0 mg), Pd(PPh₃)₄ (12.0 mg) in a 250 mL heat-resistant glass reactor. The reaction mixture was dissolved in a solvent system consisting of DMF (30.0 mL) and Et₃N (30.0 mL) under argon atmosphere. The reaction was maintained at 100 °C for 72 h under vigorous stirring in an oil bath. After cooling to room temperature, the resulting precipitate was isolated by filtration and sequentially washed with MeOH and $CH_2Cl_2$ to remove residual catalysts and byproducts. The purified products were vacuum-dried at 60 °C, yielding: BTH-COF (yellow powder, yield: 251.1 mg, 84.7%) and nitro-BTH-COF (brown powder, yield: 253.3 mg, 87.3%).

## Synthesis of NiFeHCF
First, 2.0 mmol of $K_4Fe(CN)_6 \cdot 3H_2O$ was dissolved in 100 mL of saturated KCl solution, and 1.0 mmol of $FeSO_4 \cdot 7H_2O$ and 1.0 mmol of $NiSO_4 \cdot 6H_2O$ were dissolved in 80 mL of saturated KCl solution (Ni:Fe = 1:1), then the latter solution was slowly dropped into the former solution with magnetic stirring at 60 °C for 12 h, followed by cooling to room temperature, vacuum filtration, washing with deionized water, and drying under vacuum at 80 °C for 12 h, respectively.

## Material characterizations
The crystallographic properties of the materials were analyzed using powder X-ray diffraction (XRD) on a Bruker D8 Advance diffractometer with Cu K radiation source. Morphological features and elemental composition mapping were investigated via field-emission scanning electron microscopy (SEM, Hitachi S-4800) coupled with energy-dispersive X-ray spectroscopy (EDS), complemented by transmission electron microscopy (TEM, JEM-2100). Fourier-transform infrared spectroscopy (FT-IR) was conducted on a Thermo Nicolet NEXUS instrument. Surface area and porosity parameters were derived from $N_2$ adsorption-desorption isotherms measured at −196 °C using a Micromeritics ASAP 2460 system, with specific surface area calculated

via the Brunauer–Emmett–Teller (BET) method and pore size distribution modeled through nonlocal density functional theory. Optical properties were assessed by ultraviolet-visible spectroscopy (JASCO V-750 UV-Vis spectrophotometer). Thermal degradation behavior was monitored under nitrogen atmosphere using thermogravimetric analysis (STA409 PC) at a 10 °C min⁻¹ ramp rate. Surface chemical states were probed by X-ray photoelectron spectroscopy (XPS) with Al Kα excitation. For ex-situ analyses (FT-IR, XPS, SEM, UV-Vis), cycled nitro-BTH-COF negative electrodes were retrieved from disassembled cells at designated potentials, rigorously rinsed with deionized water to remove electrolyte residues and separator fragments, and vacuum-dried at 60 °C for 24 h prior to characterization.

### Fabrication of working electrodes

The working electrodes were prepared by dispersing 70 wt% nitro-BTH-COF (active material), 20 wt% Super P (conductive additive), and 10 wt% PVDF (binder) in NMP. The slurry was stirred for 2 h to ensure homogeneity and then cast onto a carbon felt current collector using an automatic coating machine (MSK-AFA-III). The coated electrodes were dried at 80 °C for 12 h under vacuum. Subsequently, the electrodes were punched into 14 mm discs using a manual disc cutter (MSK-T10) and further dried at 80 °C overnight under vacuum prior to cell assembly. The areal mass loading of the active material was approximately 2.2 mg cm⁻².

### Cell assembly

Swagelok cells were assembled using nitro-BTH-COF as the working electrode, activated carbon as the counter electrode, and Ag/AgCl as the reference electrode. A glass fiber membrane was employed as the separator, with 100 μL 2 mol L⁻¹ $(NH_4)_2SO_4$ aqueous solution added as the electrolyte. For full-cell evaluation, CR-2032-coin cells were fabricated with NiFeHCF as the positive electrode and nitro-BTH-COF as the negative electrode. The assembly sequence consisted of the positive electrode case, NiFeHCF (5.2 mg cm⁻², single-side coated, areal capacity: 0.62 mAh cm⁻²), glass fiber separator (Whatman, 18 mm), nitro-BTH-COF (2.2 mg cm⁻², single-side coated, areal capacity: 0.70 mAh cm⁻²), spacer, spring, and the negative electrode case. The cell was designed with a negative-to-positive (N/P) capacity ratio of 1.12, and the electrolyte dosage was fixed at 100 μL of 2 mol L⁻¹ $(NH_4)_2SO_4$.

### Electrochemical tests

Cyclic voltammetry (CV) and electrochemical impedance spectroscopy (EIS) were performed using a CHI660E electrochemical workstation. For EIS measurements, a potentiostatic perturbation with an amplitude of 5 mV was applied over a frequency range of 0.01 Hz to 100 kHz (with a data density of 12 points per decade). Prior to each EIS measurement, the cells were held at open-circuit potential for 1000 s to establish a quasi-stationary state. The galvanostatic charge/discharge (GCD) measurement of $NH_4^+$-ion devices was performed on a Neware battery test system (CT-4008Tn-5V10mA-164, Shenzhen, China). All electrochemical tests were conducted at 25 ± 0.5 °C under ambient pressure. The specific capacity ($C_m$, mAh g⁻¹), specific energy ($E$, Wh kg⁻¹) and specific power ($P$, W kg⁻¹) were calculated from GCD curves using the following equations:

$$Cm = \frac{I \times \Delta t}{m} \tag{1}$$

$$E = Cm \times \Delta V \tag{2}$$

$$P = \frac{E}{1000 \times \Delta t} \tag{3}$$

where $I$ (A g⁻¹), $\Delta t$ (s), $m$ (g), $\Delta V$ (V) are the specific current, discharge time, mass loading of active substance on the elextrode and voltage window, respectively.

### Density functional theory (DFT) calculation

The theoretical calculations were conducted using the Gaussian 16 program suite. All the structures were optimized at the B3LYP-D3/def2-SVP level of theory (Supplementary Data 1–4). The electrostatic potential (ESP) was analyzed, with negative ESP regions (red) indicating electrophilic sites and positive ESP regions (blue) representing nucleophilic sites. The π-electron localization function (ELF-π) and localized orbital locator-π (LOL-π) were computed using Multiwfn 3.8 programs. The molecular orbital levels of molecules, including the highest occupied molecular orbital (HOMO) and the lowest unoccupied molecular orbital (LUMO), along with the charge population sum of nitro-BTH-COF were investigated at the B3LYP-D3/TZVP level of theory. Independent gradient model based on Hirshfeld partition (IGMH) simulations were performed with the Multiwfn program to investigate the type of interaction force. Obviously, H-bonding interactions between $NH_4^+$ and nitro-BTH-COF is revealed when the value of sign($\lambda_2$)$\rho$ approaches zero. All the DFT calculations were carried out via the Vienna Ab initio Simulation Package (VASP) and the projector augmented wave (PAW) method. The exchange-correlation functional was treated using the generalized gradient approximation (GGA) in the form of the Perdew–Burke–Ernzerhof (PBE) functional, with Grimme's D3 dispersion correction. The energy cutoff for the plane wave basis expansion was set to 400 eV. Partial occupation width of 0.2 eV was allowed for the Kohn−Sham orbitals via Gaussian smearing. The Brillouin zone was sampled using a 4×4×1 Monkhorst mesh during structural optimization. The convergence energy threshold for the self-consistent calculations was set to 10⁻⁵ eV, and the force convergency was set to 0.05 eV Å⁻¹. The uptake energy ($E_{uptake}$) of $NH_4^+$ is defined as the following form: $E_{uptake} = E(COF + nNH_4^+) - E(COF) - nE(NH_4^+)$, where $E(COF)$, $E(NH_4^+)$, and $E(COF + nNH_4^+)$ represent the energies of nitro-BTH-COF, $NH_4^+$ (excluding fragment of nitro-BTH-COF), and the total energy, respectively. A negative value of $E_{uptake}$ signifies to a stronger interaction and a more stable structure. The Gibbs free energy change ($\Delta G$) was computed using the equation $\Delta G = \Delta E + \Delta E_{ZPE} - T\Delta S$, where $\Delta E$ denotes the electronic energy difference obtained from DFT calculations. $\Delta E_{ZPE}$ is the zero-point energy correction, calculated as the difference between the adsorbed and gas-phase states. The temperature $T$ was fixed at 298.15 K to match the reaction conditions, while $\Delta S$ captures the entropy difference between the adsorbed and gas-phase species. The charge density differences were analyzed by VASPKIT code. In order to quantitatively analyze the bonding properties of $NH_4^+$ adsorbed on nitro-BTH-COF and characterization of charge transfer, the charge densities of $NH_4^+$ in the corresponding compounds were extracted from the corresponding charge densities in the nitro-BTH-COF substrates to assess the differences in the charge densities of $NH_4^+$ adsorbed on the constructed nitro-BTH-COF models. The level of charge transfer between $NH_4^+$ and nitro-BTH-COF was calculated using a Bader charge analysis program.

$$\Delta\rho = \rho(NH_4^+/nitro-BTH-COF) - \rho(nitro-BTH-COF) - \rho(NH_4^+) \tag{4}$$

### Molecular dynamic (MD) calculation

MD simulations were conducted on the Forcite module adopting the COMPASS III force field. The simulation box has a dimension of 78.55 × 78.55 × 78.55 Å³. Two amorphous solid-liquid interface systems

were constructed based on the stoichiometric ratios: a pure $(NH_4)_2SO_4$ electrolyte containing 4400 $H_2O$ molecules and $100NH_4^+/200SO_4^{2-}$ ion pairs with 5 layers of COFs. The long-range electrostatic interactions were calculated according to the Ewald method. The optimized cells were annealed for 1000 cycles within the temperature range of $300-600$ K under NPT ensemble conditions, and the configuration with the lowest energy was selected for molecular dynamics simulation. The NPT and NVT dynamics simulation was performed at 298.15 K. All the snapshots were carried out at the NPT pattern with a coupling constant of 200 ps. After that, the snapshots were performed at NVT pattern for 200 ps to obtain an equilibrium state and then date collection at NVT pattern for another 200 ps.

### Activation energy

The activation energy ($E_a$, kJ mol$^{-1}$) for the charge transfer process can be obtained using the Arrhenius equation:

$$R_{ct}^{-1} = A \exp(-E_a/RT) \tag{5}$$

where $R_{ct}$ is the charge transfer resistance ($\Omega$), $A$ is constant under a stable experimental condition, $R$ represents the gas constant (8.314 J mol$^{-1}$ K$^{-1}$), and $T$ is the temperature (K). The $\ln(R_{ct}^{-1})$ values were drawed vs. 1000/$T$, and linear fitting was executed to gather $E_a$:

$$\ln(R_{ct}^{-1}) = -E_a/RT + k \tag{6}$$

where $k$ is constant.

### Redox electron transfer number

The theoretical capacity ($C_m$, mAh g$^{-1}$) of nitro-BTH-COF was calculated based on the following form:

$$C_m = \frac{n \times F}{3.6 \times M} \tag{7}$$

The electron transfer number (n) during the coordination reaction was calculated according to the following equation:

$$n = \frac{3.6 \times C_m \times M}{F} \tag{8}$$

where $M$ is the molar mass of organic molecule (g mol$^{-1}$), and $F$ is the Faraday constant (96485 C mol$^{-1}$).

### Charge storage kinetics

The charge storage kinetics of cells were analyzed by CV curves at various scan rates. The relationship between the peak current ($i$) ans scan rate ($v$) was evaluated based on the equation:

$$i = kv^b \tag{9}$$

where $k$ and $b$ are constants. The power exponent $b$ is a crucial parameter in determining the charge storage kinetics during the redox process. The $b$-value of 0.5 and 1.0 indicate a diffusion-controlled step and a surface-governed procedure, respectively.

According to the Dunn's method, the contributions of the surface capacitive contribution and the diffusion-controlled process can be quantified by taking the following equation:

$$i = k_1 v + k_2 v^{1/2} \tag{10}$$

where $k_1$ and $k_2$ are constants, $k_1 v$ and $k_2 v^{1/2}$ represent the specific current correlated with surface fast-capacitive reaction, and the specific current due to diffusion-controlled reaction, respectively. After dividing both sides by $v^{1/2}$, the above equation is reformulated as below:

$$i/v^{1/2} = k_1 v^{1/2} + k_2 \tag{11}$$

The linear relationship between $i/v^{1/2}$ and $v^{1/2}$ can be obtained by means of a linear fit, where the slope of straight line is equal to $k_1$ and the y-intercept to $k_2$. Therefore, repeat the above steps for various voltages and scan rates to quantify the contribution of both charge storage.

## Data availability

All data that support the findings of this study are presented in the manuscript and Supplementary Information. Source data are provided with this paper.

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

## Acknowledgements

This work is financially supported by the National Natural Science Foundation of China (No. 22272118, M.L.; NO. 22172111, L.G.; and NO. 22309134, Z.S.), the Shanghai Rising-Star Program (23YF1449200, Z.S.), the Zhejiang Provincial Science and Technology Project (NO. 2022C01182, Y.L.), and the Fundamental Research Funds for the Central Universities (Z.S.).

## Author contributions

Y.C. and Z.S. conceived the idea and designed the project. L.G. and M.L. supervised the experiments and edited the paper. Y.C., Y.Q., C. H., L.M., and Y. L. performed the data processing and analysis. Y.C. and D. Z. contributed to the theoretical simulations. Y.C., Z.S., L.G. and M.L. contributed to the manuscript review. All authors discussed the results and contributed to the completion of the manuscript.

## Competing interests

The authors declare no competing interests.
