## [Peer Review File · Nature Communications]

Multi-Electron Nitrobenzothiadiazole *sp*-Conjugated-Alkynyl Covalent Organic Frameworks for Ammonium-Ion Batteries

Corresponding Author: Professor Mingxian Liu

Version 0:

Reviewer comments:

Reviewer #1

(Remarks to the Author)

Aqueous ammonium ion batteries (AIBs) have attracted considerable attention due to their high safety and rapid diffusion kinetics. A variety of electrode materials have been proposed for the AIBs, but their performance often falls short in terms of future energy storage needs. Herein, Chen et al. constructed multiple two-electron-transfer nitrobenzothiadiazole COF (nitro-BTH-COF) anode via integrating alkynyl ($-C\equiv C-$) benzenes and nitro-functionalized four-electron-acceptor benzothiadiazoles. Both theory and experiments show that by pairing with high-voltage Prussian blue (NiFeHCF) cathode, high-capacity nitro-BTH-COF anode delivers state-of-the-art battery-level energy density (86.1 Wh kg^{-1} cell) and lifespan (25,000 cycles). The obtained results are intriguing, of wide interest, of good significance and of high impact. Nevertheless, there are still some technical issues that need clarification. A revision is essential to improve the quality of the manuscript. The following points are my concerns for revising the manuscript.

(1) The geometric structures reported in Fig. 1c and 1d as well as Fig. S4 indicate that BTH-COF and nitro-BTH-COF have a unique pore (only three -NSN- groups and six -NO₂ groups) structure. However, due to the asymmetric geometric structures of linkers (BHT and BHT-nitro), it is possible to form two kinds of pore structures with six -NSN- groups and twelve -NO₂ groups. Have the authors detect the COFs with different pore structures? Why should only the COF with a unique pore structure be stable?

(2) In the geometry optimization and charge density difference calculations, the dispersion-correction should be included in the energy calculations using either PBE functional due to that they generally underestimate the van der Waals dispersive interactions. Have the authors included the dispersion-correction? Which correction method was used?

(3) As for molecular dynamics simulations, the temperature, volume (or cell size) and particle numbers should be provided in Supporting Information for others to reproduce.

(4) The equations for calculating G presented in Fig. 4b should be provided in Supporting Information for a better understanding of the results. A right arrow should be used to indicate the voltage curve in Fig. 4b for clarity and easy understanding. Definition for uptake energy of NH₄⁺ should also be given in Supporting Information.

(5) Although nitro-BTH-COF||NiFeHCF battery achieves an extraordinary cycling lifespan of 25,000 cycles with 88.91% capacity retention at 10 A g^{-1} , which is the highest value among reported NH₄⁺ full batteries, there is still 11.09% degradation of capacity. The author should analyze the reasons for the 11.09% degradation of capacity after 25,000 cycles.

(6) To obtain a full comment on the materials studied, a clear comparison in terms of electrochemical performance is needed for BTH-COF and nitro-BTH-COF. Could the authors provide a radar chart or a similarity to visually represent the trade-offs between energy density, capacity, rate performance, and cycle stability between BTH-COF and nitro-BTH-COF?

(7) Why did the authors select COFs as anode materials in their investigation? It should be pointed that in aqueous batteries, partial intercalation of protons would cause the electrode structure collapse for inorganic electrode materials such as transition metal oxides and sulfides [see: ACS appl. Energy Mater. 2023, 6, 10048–10060; Phys. Chem. Chem. Phys. 2024, 26, 323–335], resulting in a reduced capacity. This may be one of the reasons for the choice of COFs in the manuscript in terms of aqueous batteries.

(8) The atoms for which the balls in different colors stand should be explained in Fig. 3f for clarity and easy understanding.

(9) Temperature at which the N₂ uptakes were measured for the data presented in Fig. S7 should be given due to the sensitivity of adsorption/desorption isotherms to temperature. In addition, the conditions (temperature, pressure, etc.) for electrochemical performance measurements should also be provided for others to reproduce.

(10) It is hard to figure out the physical meanings if the values given in Fig. S11b. Please use the same color for all specific capacities and another same color for mass loading in the figure.

(11) Because some properties calculated from the B3LYP-D3/def2-SVP level theory while others from the PBE functional.

The data from the former theory are more reliable than those from the latter. Therefore, it would be better to exclaim the theory by which the data were calculated. For instance, in Line 128, "The LUMO-HOMO gap (ΔE) of nitro-BTN-COF (1.48 eV) is lower than BTN-COF (1.54 eV, Fig. S3)" should be corrected to "The LUMO-HOMO gap (ΔE) of nitro-BTN-COF (1.48 eV from PBE functional) is lower than BTN-COF (1.54 eV from PBE functional, Fig. S3)" or "The LUMO-HOMO gap (ΔE) of nitro-BTN-COF (1.48 eV from B3LYP-D3/def2-SVP) is lower than BTN-COF (1.54 eV from B3LYP-D3/def2-SVP, Fig. S3)".

Reviewer #2

(Remarks to the Author)

This study reports a nitro-BTH-COF material with dual active sites and validates it as an anode material for AIBs. However, there are significant contradictions between the data presented in the main text and the arguments and conclusions presented by the authors. Terms such as "ZOBs" and "TNB" appearing in the conclusions section are neither defined nor referenced anywhere in the main text. These undefined terms appear to originate from external literature, which raises concerns regarding the validity and reliability of the reported conclusions. Furthermore, the manuscript contains numerous typographical errors that significantly compromise the accuracy and readability of the content. Overall, this work lacks sufficient innovation and fails to demonstrate contributions that significantly advance the field. Therefore, this work does not meet the standards for publication in Nature Communications and is hereby rejected.

1. Significant discrepancies were noted between the findings presented in the main body of the manuscript and the conclusions drawn. The conclusion section does not accurately reflect the actual results obtained. A thorough revision of the conclusion is required to ensure consistency with the data and discussions presented earlier.

2. The authors utilized a 2 M $(\text{NH}_4)_2\text{SO}_4$ electrolyte with a pH of approximately 4.3. However, the potential contribution of hydrogen ions to the overall capacity was neither experimentally verified nor excluded in this study. Furthermore, the possible involvement of proton storage within the system was not investigated. Instead, the entire capacity was attributed solely to NH_4^+ ions without substantiating evidence. This assumption is questionable, as proton co-intercalation is a well-documented phenomenon in ammonium-ion battery systems. For example, in a 2 M NH_4BF_4 aqueous electrolyte (pH \approx 4.5), proton insertion contributes significantly to the overall capacity, even at higher pH values than the 4.3 used in this study (Advanced Functional Materials, 2025, 35(10): 2416415). Therefore, claims regarding the exclusive role of NH_4^+ storage appear insufficiently rigorous without compelling experimental validation.

3. The nitro group, as an electron-withdrawing moiety, not only lowers the HOMO energy level but also reduces the ability of the COF to lose electrons, leading to a positive shift in its redox potential. This shift is unfavorable for electrode materials intended to function as cathodes, since a positive movement in the potential window ultimately decreases the overall battery voltage. Notably, the author does not elaborate on how the introduction of the nitro group affects the redox peaks or influences the anode material, and only mentions briefly the appearance of two additional redox peaks. Such limited analysis is clearly insufficient.

4. It is noteworthy that the long-term cycling test presented in Figure 2d was performed at a high current density of 20 A g⁻¹. Although the number of cycles reached 70,000, the total test duration may not be substantially longer than that reported in studies using lower current densities (typically 6-10 A g⁻¹) with fewer cycles, due to the accelerated rate of testing. Moreover, as shown in Figure S12 (Supporting Information), at a low current density of 0.2 A g⁻¹, nitro-BTH-COF retained only 98.05% of its capacity after 5,000 cycles, while BTH-COF retained 90.12%, indicating non-negligible capacity decay even under a low current density.

5. As shown in the CV profile in Figure 2a, all three redox couples exhibit considerable peak potential separations along with markedly asymmetric shapes, suggesting poor reversibility of the redox processes in the nitro-BTH-COF.

6. In contrast to most reported studies where NH_4^+ binding leads to a reduction in both HOMO and LUMO energy levels and a narrowing of the energy gap, the present work demonstrates an increase in these energy levels upon binding. What could be the underlying reasons for this apparent discrepancy?

7. In Figure 2f, the unit for scan rate should be written in lowercase as "mV s⁻¹" (not "mV S⁻¹"). Please correct this error.

8. There is a discrepancy between the text in the Results and Discussion section (Lines 181-182) and the figure citation. The text describing the GITT data incorrectly references Figure 4h. This should be corrected to reference Figure 2h.

9. The description of the free energy hierarchy in Figure 4b (State II) contains an error. The first term in the inequality is incorrect; it is likely a typographical error. The statement " $\Delta G_1 < \Delta G_2 - 1 < \Delta G_2 - 2$ " should be revised to " $\Delta G_2 < \Delta G_2 - 1 < \Delta G_2 - 2$ " to ensure accuracy.

Reviewer #3

(Remarks to the Author)

This paper presents a comprehensive study on the development of multi-electron nitrobenzothiadiazole-based covalent organic frameworks (COFs) as advanced anode materials for high-performance aqueous ammonium-ion batteries. The authors propose incorporating multiple two-electron redox-active motifs into COFs to address the limitations of traditional single-electron reactions and significantly enhance NH_4^+ storage capacity. Additionally, the use of alkynyl-bridged sp-conjugated networks is shown to improve the framework's structural integrity and resistance to dissolution in aqueous electrolytes. The resulting nitro-BTH-COF anode demonstrates exceptional cycling stability, retaining 93.70% of its capacity after 70,000 cycles at 20 A g⁻¹, establishing a new benchmark among both organic and inorganic electrode materials. Furthermore, a full-cell configuration using nitro-BTH-COF||NiFeHCF delivers a high-power density and a state-of-the-art energy density of 86.1 Wh kg⁻¹, ranking among the highest reported for NH_4^+ full batteries to date.

Overall, the study is comprehensive, well-structured, and supported by extensive experimental and theoretical techniques, including the independent gradient model (IGM) and differential charge density isosurfaces. However, the conclusion section appears somewhat disconnected from the main content of the manuscript and requires a thorough revision. A few other suggestions for improvement are outlined below.

Comments:

1. The authors have employed Sonogashira coupling to construct the COF. This synthetic strategy is infrequently used in COF development, although it has been previously applied in the synthesis of covalent triazine-based frameworks (CTFs). Could the authors elaborate on the rationale behind choosing this triple-bond-forming linkage over more commonly reported CC bond-forming strategies in COF chemistry? A discussion on the advantages or intended design benefits of this approach would strengthen the justification for its selection.
2. 2- It would be helpful to include a comparison of the PXRD pattern with those of the starting monomers in the Supporting Information to illustrate the framework formation better. 3- (Line 139):
3. Please revise the FTIR peak assignments or elaborate. For example, assigning the peak at 1570 cm^{-1} , as $\text{C}\equiv\text{C}$ stretching typically appears around $2100\text{--}2200\text{ cm}^{-1}$. The $1570\text{--}1600\text{ cm}^{-1}$ region is more commonly associated with $\text{C}=\text{C}$ stretching vibrations, particularly in aromatic systems. 4-(Line 29):
4. "Imine bonds" might be more relevant, as they better represent dynamic covalent chemistry than "amine bonds."
5. Please revise the conclusion, as it seems irrelevant to the current manuscript.
6. The equations show that energy density (E) is calculated using the discharge time, current, and the mass of the active substance on a single electrode, rather than the total mass of both electrodes in the full cell. As a result, the reported 86.1 Wh kg^{-1} likely represents an anode-based specific energy, not a true full-cell value. The authors should clarify the mass basis used for these calculations and, if applicable, recalculate the energy density considering the total active material of both electrodes for a fair comparison with other full-cell systems.
7. Please include the X-ray Photoelectron Spectroscopy (XPS) analysis for the synthesized BTH-COF material. This characterization is essential to confirm the elemental composition and chemical states of the constituent atoms. In particular, XPS spectra should be used to identify bonding interactions and the chemical environment before and after the TEB modification step. Please provide a detailed deconvolution of the relevant peaks to demonstrate the successful modification and to clarify the nature of the new chemical bonds (N-O).
8. The authors report high b-values ($0.91\text{--}0.96$, Fig. 2g), indicating a surface-controlled pseudocapacitive process. However, the analysis does not quantitatively separate capacitive and diffusion-controlled contributions. A further capacitive contribution analysis using the relation at various scan rates would provide a clearer understanding of the charge-storage mechanism and strengthen the electrochemical interpretation.
9. The text and labels in Figure 3b are not clearly readable. Please improve the resolution, colour, and font size of the figure and all annotations.

Version 1:

Reviewer comments:

Reviewer #1

(Remarks to the Author)

The authors have well-addressed all my comments and revised their manuscript accordingly. The authors have studied the COFs with different pore structures and demonstrated that BTH-COF and nitro-BTH-COF have a unique pore (only three -NSN- groups and six -NO₂ groups) structure in the revised version of the manuscript. They have included Grimme's D3 dispersion correction (IVDW=11) to avoid the underestimation of van der Waals interactions by the PBE functional for the geometry optimization and charge density difference calculations in the revised manuscript. Details for molecular dynamics simulations have been provided in the revised Supplementary Information, Section S1. Supplementary Methods. The authors have analyzed the reasons for the 11.09% degradation of capacity after 25,000 cycles in the revised version of the manuscript. Now the revised manuscript has been improved adequately. Considering that the manuscript is of great significance to the energy storage and conversion as well as other related fields, I would like to recommend the current form of the revised manuscript for publication.

Reviewer #2

(Remarks to the Author)

I think the current version can be published in Nature Communications.

Reviewer #3

(Remarks to the Author)

The authors have addressed the comments/concerns raised by me during the review process. I support the publication of this work in its current form.

Point-to-Point Response to the Reviewers' Comments

Response to the Comments of Reviewer 1:

We would like to express our sincere thanks to the expert reviewer for the constructive and positive comments, which significantly contributed to improving the quality of the manuscript. Please find below a detailed response to each of the comments.

Reviewer 1: Aqueous ammonium ion batteries (AIBs) have attracted considerable attention due to their high safety and rapid diffusion kinetics. A variety of electrode materials have been proposed for the AIBs, but their performance often falls short in terms of future energy storage needs. Herein, Chen et al. constructed multiple two-electron-transfer nitrobenzothiadiazole COF (nitro-BTH-COF) anode *via* integrating alkynyl ($-C\equiv C-$) benzenes and nitro-functionalized four-electron-acceptor benzothiadiazoles. Both theory and experiments show that by pairing with high-voltage Prussian blue (NiFeHCF) cathode, high-capacity nitro-BTH-COF anode delivers state-of-the-art battery-level energy density ($86.1 \text{ Wh kg}^{-1}_{\text{cell}}$) and lifespan (25,000 cycles). The obtained results are intriguing, of wide interest, of good significance and of high impact. Nevertheless, there are still some technical issues that need clarification. A revision is essential to improve the quality of the manuscript.

1. The geometric structures reported in Fig. 1c and 1d as well as Fig. S4 indicate that nitro-BTH-COF and nitro-BTH-COF have a unique pore (only three -NSN- groups and six -NO₂ groups) structure. However, due to the asymmetric geometric structures of linkers (BTH and nitro-BTH), it is possible to form two kinds of pore structures with six -NSN- groups and twelve -NO₂ groups. Have the authors detected the COFs with different pore structures? Why should only the COF with a unique pore structure be stable?

Response: Thank you for your insightful comments. We have studied this point with supplementary PXRD pattern simulations in Supplementary Fig. 4, and added related discussion in the revised Manuscript: The experimental PXRD patterns of both BTH-COF and nitro-BTH-COF match the simulated AA-stacking models (rather than AB-stacking models) in terms of peak positions and relative intensities (Fig. 1c, d and Supplementary Fig. 4). Furthermore, the experimental pore sizes of BTH-COF (1.90 nm) and nitro-BTH-COF (1.72 nm) analyzed by nitrogen adsorption/desorption

are consistent with those calculated by the AA-stacking model (1.81 and 1.67 nm, Supplementary Fig. 5). These results indicate that BTH-COF and nitro-BTH-COF have a unique pore (only three -NSN- groups and six -NO₂ groups) structure.

Supplementary Fig. 4 Experimental PXRD patterns and simulated crystal structural models of (a–c) BTH-COF and (d–f) nitro-BTH-COF.

Supplementary Fig. 5 Nitrogen adsorption/desorption isotherms and simulated pore-size distribution models of (a–c) BTH-COF and (d–f) nitro-BTH COF.

Besides, we studied the structural stability of BTH-COF and nitro-BTH-COF by ultraviolet-visible (UV-Vis) spectrum analysis in Supplementary Fig. 8, and added related discussion behind

Supplementary Fig. 8 in the revised Supplementary Information: Compared with soluble small molecules of TEB, BTH and BTH-NO₂ (Supplementary Fig. 8a), there is no UV-vis absorption signal for both BTH-COF and nitro-BTH-COF after soaking 2 M (NH₄)₂SO₄ aqueous electrolyte, confirming their structural robustness and anti-dissolution ability. The rigid alkynyl-bridged *sp*-conjugated frameworks of both BTH-COF and nitro-BTH-COF with a unique pore structure contribute to their structural stability, which is beneficial for sustained electrochemical activity.

Supplementary Fig. 8 (a) UV/Vis spectra of BTH-COF and nitro-BTH-COF soaked in 2 M (NH₄)₂SO₄ aqueous electrolyte.

2. In the geometry optimization and charge density difference calculations, the dispersion-correction should be included in the energy calculations using either PBE functional due to that they generally underestimate the van der Waals dispersive interactions. Have the authors included the dispersion-correction? Which correction method was used?

Response: Thank you for your constructive comment. We have included Grimme's D3 dispersion correction (IVDW=11) to avoid the underestimation of van der Waals interactions by the PBE functional for the geometry optimization and charge density difference calculations. We have clarified this point in the revised Supplementary Information, Section S1. Supplementary Methods: The exchange-correlation functional was treated using the generalized gradient approximation (GGA) in the form of the Perdew-Burke-Ernzerhof (PBE) functional, with Grimme's D3 dispersion correction^[S6].

3. As for molecular dynamics simulations, the temperature, volume (or cell size) and particle numbers should be provided in Supporting Information for others to reproduce.

Response: Thank you for your valuable comment. We have provided these informations in the revised Supplementary Information, Section S1. Supplementary Methods: The simulation box has

a dimension of $78.55 \times 78.55 \times 78.55 \text{ \AA}^3$. Two amorphous solid-liquid interface systems were constructed based on the stoichiometric ratios: a pure $(\text{NH}_4)_2\text{SO}_4$ electrolyte containing 4400 H_2O molecules and $100\text{NH}_4^+/200\text{SO}_4^{2-}$ ion pairs with 5 layers of COFs. The long-range electrostatic interactions were calculated according to the Ewald method. The optimized cells were annealed for 1000 cycles within the temperature range of 300–600 K under NPT ensemble conditions, and the configuration with the lowest energy was selected for molecular dynamics simulation. The NPT and NVT dynamics simulation was performed at 298.15 K.

4. The equations for calculating ΔG presented in Fig. 4b should be provided in Supporting Information for a better understanding of the results. A right arrow should be used to indicate the voltage curve in Fig. 4b for clarity and easy understanding. Definition for uptake energy of NH_4^+ should also be given in Supporting Information.

Response: Thank you for your comment. We have included the complete equations for calculating ΔG in Fig. 4b in the revised Supplementary Information, Section S1. Supplementary Methods: The Gibbs free energy change (ΔG) was computed using the equation $\Delta G = \Delta E + \Delta E_{\text{ZPE}} - T\Delta S$, where ΔE denotes the electronic energy difference obtained from DFT calculations. ΔE_{ZPE} is the zero-point energy correction, calculated as the difference between the adsorbed and gas-phase states. The temperature T was fixed at 298.15 K to match the reaction conditions, while ΔS captures the entropy difference between the adsorbed and gas-phase species.

In the revised Manuscript, a right arrow has been added to Fig. 4b to indicate the voltage curve for clarity and easy understanding.

Fig. 4 (b) Calculated ΔG values of possible reaction paths.

Besides, we have gave definition for uptake energy of NH_4^+ in the revised Supplementary Information, Section S1. Supplementary Methods: The uptake energy (E_{uptake}) of NH_4^+ is defined

as the following form: $E_{\text{uptake}} = E(\text{COF} + n\text{NH}_4^+) - E(\text{COF}) - nE(\text{NH}_4^+)$, where $E(\text{COF})$, $E(\text{NH}_4^+)$, and $E(\text{COF} + n\text{NH}_4^+)$ represent the energies of nitro-BTH-COF, NH_4^+ (excluding fragment of nitro-BTH-COF), and the total energy, respectively. A negative value of E_{uptake} signifies to a stronger interaction and a more stable structure.

5. Although nitro-BTH-COF||NiFeHCF battery achieves an extraordinary cycling lifespan of 25,000 cycles with 88.91% capacity retention at 10 A g⁻¹, which is the highest value among reported NH_4^+ full batteries, there is still 11.09% degradation of capacity. The author should analyze the reasons for the 11.09% degradation of capacity after 25,000 cycles.

Response: Thank you for your comment. We have analyzed the reasons in the revised Manuscript: XRD patterns of inorganic NiFeHCF cathode show the gradually weakened crystal peaks of (420), (422), and (620) planes (Supplementary Fig. 21a), indicating its destroyed crystalline structure during the long-term NH_4^+ -insertion/extraction cycling process. Considering the superior structure stability of nitro-BTH-COF (Supplementary Fig. 21b), the 11.09% capacity loss of nitro-BTH-COF||NiFeHCF battery can be attributed to the crystal structural degradation of NiFeHCF cathode after 25,000 cycles.

Supplementary Fig. 21 XRD patterns of (a) NiFeHCF cathode and (b) nitro-BTH-COF anode before and after cycles.

6. To obtain a full comment on the materials studied, a clear comparison in terms of electrochemical performance is needed for BTH-COF and nitro-BTH-COF. Could the authors provide a radar chart or a similarity to visually represent the trade-offs between energy density, capacity, rate performance, and cycle stability between BTH-COF and nitro-BTH-COF?

Response: Thank you for your comment. We have provided a radar chart to visually represent the

trade-offs between energy density, capacity, rate performance, and cycle stability between BTH-COF and nitro-BTH-COF in Supplementary Fig. 12d in the revised Supplementary Information, and added related discussion in the revised Manuscript: Overall, thanks to the high degree of π -electron *sp*-conjugation along alkynyl linkages and strong electron-drawing effect of nitrobenzothiadiazole motifs, nitro-BTH-COF anode liberates better comprehensive performance in terms of energy density, capacity, rate performance, and cycle stability than that of BTH-COF (Supplementary Fig. 12d), holding desirable electrochemical potential for propelling AIBs.

Supplementary Fig. 12 (d) Radar diagram of overall comparison of electrochemical performance between nitro-BTH-COF and BTH-COF.

7. Why did the authors select COFs as anode materials in their investigation? It should be pointed that in aqueous batteries, partial intercalation of protons would cause the electrode structure collapse for inorganic electrode materials such as transition metal oxides and sulfides [see: ACS appl. Energy Mater. 2023, 6, 10048–10060; Phys. Chem. Chem. Phys. 2024, 26, 323–335], resulting in a reduced capacity. This may be one of the reasons for the choice of COFs in the manuscript in terms of aqueous batteries.

Response: Thank you for your constructive comments. We have clarified the reason of selecting COFs as anode materials in electrochemical investigation in the revised Manuscript, Introduction: Covalent organic frameworks (COFs), as crystalline porous polymers formed *via* periodically covalent linkage of π -conjugated building blocks, offer a unique structural platform to address the activity and stability limitations of organic molecules, uncontrolled polymers, and inevitable structure collapse of inorganic materials with ionic intercalation⁴¹⁻⁴⁷. Their ordered architectures enable precise spatial organization of redox units, controlled pore environments, and enhanced

charge transfer kinetics, making them ideal candidates for NH_4^+ storage⁴⁸⁻⁵⁰. Here some representative studies including the mentioned works were provided to broaden the research background of the aqueous batteries.

8. The atoms for which the balls in different colors stand should be explained in Fig. 3f for clarity and easy understanding.

Response: Thank you for your valuable suggestion. We have explained the atoms for the balls in different colors in Fig. 3f for clarity and easy understanding in the revised Manuscript:

Fig. 3 (f) H-bonding NH_4^+ coordination mechanism.

9. Temperature at which the N_2 uptakes were measured for the data presented in Fig. S7 should be given due to the sensitivity of adsorption/desorption isotherms to temperature. In addition, the conditions (temperature, pressure, etc.) for electrochemical performance measurements should also be provided for others to reproduce.

Response: Thank you for your constructive comment. We have provided these informations in the revised Manuscript, Method: Surface area and porosity parameters were derived from N_2 adsorption-desorption isotherms measured at $-196\text{ }^\circ\text{C}$ using a Micromeritics ASAP 2460 system. All electrochemical tests were conducted at room temperature ($25\text{ }^\circ\text{C}$) under ambient pressure.

10. It is hard to figure out the physical meanings if the values given in Fig. S11b. Please use the same color for all specific capacities and another same color for mass loading in the figure.

Response: Thank you for your valuable suggestion. To clearly describe the variation of specific capacities of nitro-BTH-COF anode at different mass loadings, we have used the same color of "blue" for all specific capacities, and another same color of "green" for mass loading in Supplementary Fig. 11d in the revised Supplementary Information.

Supplementary Fig. 11 (d) Capacities of nitro-BTH-COF anode at different mass loadings.

11. Because some properties calculated from the B3LYP-D3/def2-SVP level theory while others from the PBE functional. The data from the former theory are more reliable than those from the latter. Therefore, it would be better to explain the theory by which the data were calculated. For instance, in Line 128, "The LUMO-HOMO gap (ΔE) of nitro-BTN-COF (1.48 eV) is lower than BTN-COF (1.54 eV, Fig. S3)" should be corrected to "The LUMO-HOMO gap (ΔE) of nitro-BTN-COF (1.48 eV from PBE functional) is lower than BTN-COF (1.54 eV from PBE functional, Fig. S3)" or "The LUMO-HOMO gap (ΔE) of nitro-BTN-COF (1.48 eV from B3LYP-D3/def2-SVP) is lower than BTN-COF (1.54 eV from B3LYP-D3/def2-SVP, Fig. S3)".

Response: Thank you for your insightful comments. We have claimed this point in revised Manuscript: The LUMO-HOMO gap (ΔE) of nitro-BTN-COF (1.48 eV from B3LYP-D3/TZVP) is lower than BTN-COF (1.54 eV from B3LYP-D3/TZVP, Supplementary Fig. 3), promising superior electron transfer efficiency. Frontier molecular orbital analysis reveals charge-storage intermediates possess ideal electronic properties for NH_4^+ storage, exhibiting both strong electron affinity (progressively increasing LUMO levels) and excellent charge transfer capability (narrow HOMO-LUMO gaps of 0.58, 0.66, 0.71 eV from B3LYP-D3/TZVP, Fig. 4f).

Response to the Comments of Reviewer 2:

We would like to express our sincere thanks to the expert reviewer for the constructive and positive comments, which significantly contributed to improving the quality of the manuscript. Please find below a detailed response to each of the comments.

Reviewer 2: This study reports a nitro-BTH-COF material with dual active sites and validates it as an anode material for AIBs. **a)** However, there are significant contradictions between the data presented in the main text and the arguments and conclusions presented by the authors. Terms such

as "ZOBs" and "TNB" appearing in the conclusions section are neither defined nor referenced anywhere in the main text. These undefined terms appear to originate from external literature, which raises concerns regarding the validity and reliability of the reported conclusions. Furthermore, the manuscript contains numerous typographical errors that significantly compromise the accuracy and readability of the content. **b)** Overall, this work lacks sufficient innovation and fails to demonstrate contributions that significantly advance the field. Therefore, this work does not meet the standards for publication in Nature Communications and is hereby rejected.

Response: Thank you for your insightful comments. We have categorized the Reviewer's questions into a), and b).

a) We sincerely apologize for the incorrect demonstration of the data in the conclusion section, which results in a contradiction between the data presented in the main text and the arguments and conclusions. We have carefully revised the Conclusion in the revised Manuscript: In conclusion, multiple two-electron **nitro-BTH-COF** is designed by fusing alkynyl benzenes and nitrobenzothiadiazole units. **nitro-BTH-COF** delivers highly π -electron *sp*-conjugation along alkynyl linkages and strong electron-drawing nitrobenzothiadiazole motifs. **This feature promises high NH_4^+ utilization (95.2%) of multi-two-electron nitro/thiazole sites with a lower activation energy (25.93 vs. 35.99 kJ mol^{-1} of BTH-COF).** Experiment and theoretical studies reveal the fast and stable **octadeca-H-bonded NH_4^+** coordination mechanism of **nitro-BTH-COF anode**. As a consequence, **nitro-BTH-COF anode** liberates the highest capacity among reported COFs in **AIBs**. Furthermore, the alkynyl-bridged π -conjugation network of **nitro-BTH-COF anode** gives excellent structural insolubility in aqueous electrolyte, affording unprecedented cycling life. By pairing **nitro-BTH-COF anode** with high-voltage Prussian blue cathode, the constructed **nitro-BTH-COF||NiFeHCF** full battery shows state-of-the-art energy density and cycling life. This work gives new insights into the design of multi-electron-redox and stable COFs for advanced **AIBs**.

Besides, we have revised the typographical errors including **Fig. 2f, h, and Fig. 4b** in the main text, and carefully checked the Manuscript to ensure the accuracy and readability of the content.

b) Regarding the innovation issue, there have been a few preliminary investigations on COFs anodes for aqueous ammonium-ion batteries (AIBs), such as quinone-pyrazine COFs and super-conjugated amine-linked anthraquinone COFs. These achievements broaden the design horizon of COFs to boost the capacity of AIBs, but generally requires the introduction of substantial single-

electron active motifs (*e.g.*, C=O, C=N) based on dynamic amine bond coupling. Unfortunately, this strategy has almost reached the capacity saturation point of COFs (<250 mAh g⁻¹) under limited redox efficiency, meanwhile it brings structural instability caused by twisted –NH– linkages, resulting in unsatisfactory cycling life (<10,000 cycles). Overall, the single-electron site reactions and twisted instable linkages of current COFs anodes pose significant challenges for advancing AIBs.

Different from these reported works, we unlock multi-electron-transfer nitrobenzothiadiazole *sp*-conjugated-alkynyl COF anode, which breaks the efficiency limitation of single-electron site reactions and the twisted instable linkages, enabling state-of-the-art aqueous AIBs. The novelty and the significance of this work are:

(1) nitro-BTH-COF is designed by integrating robust alkynyl (–C≡C–) benzenes and four-electron-acceptor nitrobenzothiadiazoles, showing high degree of π -electron *sp*-conjugation along alkynyl linkages, strong electron-drawing effect of nitrobenzothiadiazole motifs, and excellent structural anti-dissolution in aqueous electrolyte.

(2) nitro-BTH-COF anode promises high NH₄⁺-accessibility of multi-two-electron nitro/thiazole motifs (95.2% utilization) with a lower activation energy (25.93 *vs.* 35.99 kJ mol⁻¹ of BTH-COF), thereby liberating the highest capacity (317 mA h g⁻¹) and cycling stability (70,000 cycles) among all reported COFs via fast and stable 12 e⁻ H-bonded NH₄⁺ coordination mechanism.

(3) By pairing with high-voltage Prussian blue (NiFeHCF) cathode, high-capacity nitro-BTH-COF anode delivers state-of-the-art battery-level energy density (86.1 Wh kg⁻¹_{cell}) and lifespan (25,000 cycles), highlighting its great promise for advanced AIBs.

As the Reviewer 1 has commented: "The obtained results are intriguing, of wide interest, of good significance and of high impact.", we believe our work marks significant progress in designing multi-electron and stable *sp*-conjugated-alkynyl COFs, which will inspire further efforts to enrich the organic structure library in the energy community.

1. Significant discrepancies were noted between the findings presented in the main body of the manuscript and the conclusions drawn. The conclusion section does not accurately reflect the actual results obtained. A thorough revision of the conclusion is required to ensure consistency with the data and discussions presented earlier.

Response: Thank you for your constructive comment. In the revised Manuscript, we have provided a thorough revision of the conclusion to ensure consistency with the data and discussions presented earlier: In conclusion, multiple two-electron nitro-BTH-COF is designed by fusing alkynyl benzenes and nitrobenzothiadiazole units. nitro-BTH-COF delivers highly π -electron *sp*-conjugation along alkynyl linkages and strong electron-drawing nitrobenzothiadiazole motifs. This feature promises high NH_4^+ utilization (95.2%) of multi-two-electron nitro/thiazole sites with a lower activation energy (25.93 vs. 35.99 kJ mol^{-1} of BTH-COF). Experiment and theoretical studies reveal the fast and stable octadeca-H-bonded NH_4^+ coordination mechanism of nitro-BTH-COF anode. As a consequence, nitro-BTH-COF anode liberates the highest capacity among reported COFs in AIBs. Furthermore, the alkynyl-bridged π -conjugation network of nitro-BTH-COF anode gives excellent structural insolubility in aqueous electrolyte, affording unprecedented cycling life. By pairing nitro-BTH-COF anode with high-voltage Prussian blue cathode, the constructed nitro-BTH-COF||NiFeHCF full battery shows state-of-the-art energy density and cycling life. This work gives new insights into the design of multi-electron-redox and stable COFs for advanced AIBs.

2. The authors utilized a 2 M $(\text{NH}_4)_2\text{SO}_4$ electrolyte with a pH of approximately 4.3. However, the potential contribution of hydrogen ions to the overall capacity was neither experimentally verified nor excluded in this study. Furthermore, the possible involvement of proton storage within the system was not investigated. Instead, the entire capacity was attributed solely to NH_4^+ ions without substantiating evidence. This assumption is questionable, as proton co-intercalation is a well-documented phenomenon in ammonium-ion battery systems. For example, in a 2 M NH_4BF_4 aqueous electrolyte ($\text{pH}\approx 4.5$), proton insertion contributes significantly to the overall capacity, even at higher pH values than the 4.3 used in this study (*Advanced Functional Materials*, **2025**, 35(10): 2416415). Therefore, claims regarding the exclusive role of NH_4^+ storage appear insufficiently rigorous without compelling experimental validation.

Response: Thank you for your insightful comments. We have studied the possible H^+ involvement during NH_4^+ storage in nitro-BTH-COF anode in Supplementary Figs. 10 and 11. Related discussion was added behind Supplementary Fig. 11 in revised Supplementary Information: Considering the weak acidity of 2 M $(\text{NH}_4)_2\text{SO}_4/\text{H}_2\text{O}$ electrolyte ($\text{pH}=4.90$) and the small size of H^+ ions, the contribution of H^+ ion to the total capacity storage of nitro-BTH-COF anode was

studied in $\text{H}_2\text{SO}_4/\text{H}_2\text{O}$ electrolyte (with the same pH value as 2 M $(\text{NH}_4)_2\text{SO}_4/\text{H}_2\text{O}$ electrolyte). As previously reported^[S11], H^+ co-intercalation is a well-documented phenomenon in AIBs. In our case, nitro-BTH-COF anode in $\text{H}_2\text{SO}_4/\text{H}_2\text{O}$ electrolyte exhibits completely different redox peaks and charge storage behaviors (Supplementary Fig. 10e) with a negligible capacity of 7.8 mAh g^{-1} (Supplementary Fig. 10f), far below the 317 mAh g^{-1} observed in $(\text{NH}_4)_2\text{SO}_4/\text{H}_2\text{O}$ electrolyte. This result excludes H^+ involvement in the redox process of nitro-BTH-COF anode. Furthermore, we investigated the high-resolution N 1s and O 1s XPS spectra of nitro-BTH-COF anode in $\text{H}_2\text{SO}_4/\text{H}_2\text{O}$ electrolyte at different electrochemical states. During the electrochemical process, the C=N (398.9 eV) and $-\text{NO}_2$ groups (405.6/533.5 eV) of nitro-BTH-COF anode remain unchanged without significant shifts or impure signal formation caused by H^+ reaction (Supplementary Fig. 11a and b). These results confirm that the role of H^+ ions can be ignored, and NH_4^+ ions are primarily responsible for the capacity storage of nitro-BTH-COF anode.

Supplementary Fig. 10 Electrochemical performance of nitro-BTH-COF at different electrolyte system: (e) CV curves at 0.1 mV s^{-1} , (f) GCD profiles at 0.2 A g^{-1} .

Supplementary Fig. 11 High-resolution (a) N 1s and (b) O 1s XPS spectra of nitro-BTH-COF anode in $\text{H}_2\text{SO}_4/\text{H}_2\text{O}$ electrolyte (pH=4.90) during the electrochemical process.

3. The nitro group, as an electron-withdrawing moiety, not only lowers the HOMO energy level

but also reduces the ability of the COF to lose electrons, leading to a positive shift in its redox potential. This shift is unfavorable for electrode materials intended to function as cathodes, since a positive movement in the potential window ultimately decreases the overall battery voltage. Notably, the author does not elaborate on how the introduction of the nitro group affects the redox peaks or influences the anode material, and only mentions briefly the appearance of two additional redox peaks. Such limited analysis is clearly insufficient.

Response: Thank you for your insightful comments. We have provided a clearly analysis about this point behind Supplementary Fig. 10 in the revised Supplementary Information: nitro-BTH-COF was served as the anode (rather than the cathode) in AIBs. Electron-withdrawing nitro groups act as electron acceptors to easily undergo reduction reaction to couple NH_4^+ ions, which are dependent of the LUMO energy level. Indeed, the introduction of nitro groups lowers the LUMO energy level from -7.72 (BTH-COF) to -7.87 eV (nitro-BTH-COF, Supplementary Fig. 3), resulting in a slight positive shift in the average redox potential from -0.56 V (BTH-COF, Supplementary Fig. 10d) to -0.49 V (nitro-BTH-COF, Fig. 2b). Of note, with the ability to accept two electrons per nitro group, it can trigger extra multielectron redox reactions for nitro-BTH-COF anode to achieve a superior capacity of 317 mAh g^{-1} compared to BTH-COF anode (117 mAh g^{-1} , Supplementary Fig. 10d). Overall, nitro groups of high-capacity nitro-BTH-COF anode do not significantly damage the voltage (1.1 V) of nitro-BTH-COF||NiFeHCF full battery, which thus liberates state-of-the-art battery-level energy density ($86.1 \text{ Wh kg}^{-1}_{\text{cell}}$) among all reported NH_4^+ full batteries (Fig. 5f and Supplementary Table 2).

4. It is noteworthy that the long-term cycling test presented in Figure 2d was performed at a high current density of 20 A g^{-1} . Although the number of cycles reached 70,000, the total test duration may not be substantially longer than that reported in studies using lower current densities (typically $6\text{--}10 \text{ A g}^{-1}$) with fewer cycles, due to the accelerated rate of testing. Moreover, as shown in Figure S12 (Supporting Information), at a low current density of 0.2 A g^{-1} , nitro-BTH-COF retained only 98.05% of its capacity after 5,000 cycles, while BTH-COF retained 90.12%, indicating non-negligible capacity decay even under a low current density.

Response: Thank you for your valuable suggestion. To confirm the electrochemical stability of nitro-BTH-COF anode at lower current densities, we further supplemented its cyclic performance

at 6 A g^{-1} in Supplementary Fig. 12c, and added the related discussion in revised Manuscript: In addition, nitro-BTH-COF anode at 6 A g^{-1} delivers a high-capacity retention of 94.59% over 20,000 cycles (Supplementary Fig. 12c), showing its desirable electrochemical stability.

Supplementary Fig. 12 (c) Cycling performance of nitro-BTH-COF anode 6 A g^{-1} .

In Supplementary Fig. 12, BTH-COF anode exhibits non-negligible capacity fading after 5000 cycles (9.88% capacity loss: 12 mAh g^{-1}). This degradation is ascribed to the attenuated activity and reversibility of thiazole active sites (*Angew. Chem. Int. Ed.* **2023**, *62*, e202216136; Ref. 2). In contrast, nitro-BTH-COF anode shows only 1.95% capacity loss (5.4 mAh g^{-1}) after 5000 cycles. This is because the induction of strong electron-drawing nitro groups in nitro-BTH-COF effectively promise high NH_4^+ accessibility of multi-two-electron nitro/thiazole sites of 95.2% utilization with low activation energy of $25.93 \text{ kJ mol}^{-1}$ (vs. 47.5% utilization/ $35.99 \text{ kJ mol}^{-1}$ of BTH-COF). Thus, nitro-BTH-COF anode at 0.2 A g^{-1} achieves better capacity retention over 5000 cycles than that of BTH-COF (Supplementary Fig. 12a and b).

Besides, we explained this point in the revised Manuscript: nitro-BTH-COF anode at 0.2 A g^{-1} achieves better capacity retention over 5000 cycles than that of BTH-COF (Supplementary Fig. 12a and b), due to the attenuated activity and reversibility of thiazole active sites².

5. As shown in the CV profile in Figure 2a, all three redox couples exhibit considerable peak potential separations along with markedly asymmetric shapes, suggesting poor reversibility of the redox processes in the nitro-BTH-COF.

Response: Thank you for your insightful comments. It should be noted that both the oxidation and reduction process of nitro-BTH-COF anode show three redox signals (Fig. 2a), implying that the NH_4^+ redox reaction is reversible. Meanwhile, the capacities as a function of current densities

within successive five cycles can reversibly revert back to the original levels upon switching the current density to 0.2 A g^{-1} (Fig. 2c), confirming the desirable electrochemical process with high reversibility. This can be further reflected by the 100% Coulombic efficiency of nitro-BTH-COF anode after long-term 70,000 cycles (Fig. 2d). DFT calculations also confirm the highly reversible NH_4^+ reaction process (Fig. 4a and b). These results suggest that, despite slight peak potential separations and asymmetric shapes, nitro-BTH-COF anode still undergoes a reversible multi-electron redox process at nitro/thiazole motifs, where the three observed redox couples represent partially overlapping signals from sequential electron-transfer steps, each step involving reduction of electron-deficient sites coupled with NH_4^+ for charge balance.

Besides, the potential separations and asymmetric shapes of CV profiles have been well-documented in recently reported redox-active organic materials, which do not significantly affect their electrochemical reversibility and charge storage performances (*e.g.*, *Angew. Chem. Int. Ed.* **2025**, 64, e202424494, Ref.51; *Adv. Mater.* **2024**, 36, 2409354, Ref.43; *Adv. Mater.* **2024**, 36, 2308210, Ref.30; *Nat. Commun.* **2021**, 12, 4424).

6. In contrast to most reported studies where NH_4^+ binding leads to a reduction in both HOMO and LUMO energy levels and a narrowing of the energy gap, the present work demonstrates an increase in these energy levels upon binding. What could be the underlying reasons for this apparent discrepancy?

Response: Thank you for your insightful comments. We have clarified this point in the revised Manuscript: NH_4^+ binding in nitro-BTH-COF via multiple H-bonding interactions ($\text{N-H}\cdots\text{O}$ and $\text{N-H}\cdots\text{N}$) introduces increased steric congestion that restricts structural relaxation^{41,70,71}. This drives an upward shift in both HOMO and LUMO energy levels and results in a modest widening of the energy gap (from 0.58 to 0.71 eV). Thus, the energy level upshift is a typical feature upon ion binding, which also has been well-documented in recently reported organic electrode materials, *e.g.*, *J. Am. Chem. Soc.* **2025**, 147, 40, 36547 (Ref.71); *Angew. Chem. Int. Ed.* **2025**, 64, e202501743; *Angew. Chem. Int. Ed.* **2025**, 64, e202507570 (Ref.70); *Angew. Chem. Int. Ed.* **2023**, 62, e202309446 (Ref.41).

7. In Figure 2f, the unit for scan rate should be written in lowercase as " mV s^{-1} " (not " mV S^{-1} "). Please correct this error.

Response: Thank you for your careful review. We have corrected the unit of "mV S⁻¹" into "mV s⁻¹" for the scan rate in Fig. 2f in the revised Manuscript.

8. There is a discrepancy between the text in the Results and Discussion section (Lines 181-182) and the figure citation. The text describing the GITT data incorrectly references Figure 4h. This should be corrected to reference Figure 2h.

Response: Thank you for your careful review. We have corrected the figure citation in the Results and Discussion section from Fig. 4h to Fig. 2h in the revised Manuscript: Galvanostatic intermittent titration technique (GITT) analysis further confirms the high NH₄⁺ diffusion coefficient of 10⁻⁸~10⁻⁷ cm² s⁻¹ for nitro-BTH-COF anode (Fig. 2h and Supplementary Fig. 4c), surpassing previously reported values in organic/inorganic hosts (~10⁻¹²–10⁻⁸).

9. The description of the free energy hierarchy in Figure 4b (State II) contains an error. The first term in the inequality is incorrect; it is likely a typographical error. The statement " $\Delta G_1 < \Delta G_{2-1} < \Delta G_{2-2}$ " should be revised to " $\Delta G_2 < \Delta G_{2-1} < \Delta G_{2-2}$ " to ensure accuracy.

Response: Thank you for your careful review. We have revised the inequality in Figure 4b (State II) from " $\Delta G_1 < \Delta G_{2-1} < \Delta G_{2-2}$ " to " $\Delta G_2 < \Delta G_{2-1} < \Delta G_{2-2}$ " to ensure accuracy and consistency with the underlying calculations.

Fig. 4 (b) Calculated ΔG values of possible reaction paths.

Response to the Comments of Reviewer 3:

We would like to express our sincere thanks to the expert reviewer for the constructive and positive comments, which significantly contributed to improving the quality of the manuscript. Please find below a detailed response to each of the comments.

Reviewer 3: This paper presents a comprehensive study on the development of multi-electron nitrobenzothiadiazole-based covalent organic frameworks (COFs) as advanced anode materials for high-performance aqueous ammonium-ion batteries. The authors propose incorporating multiple two-electron redox-active motifs into COFs to address the limitations of traditional single-electron reactions and significantly enhance NH_4^+ storage capacity. Additionally, the use of alkynyl-bridged *sp*-conjugated networks is shown to improve the framework's structural integrity and resistance to dissolution in aqueous electrolytes. The resulting nitro-BTH-COF anode demonstrates exceptional cycling stability, retaining 93.70% of its capacity after 70,000 cycles at 20 A g^{-1} , establishing a new benchmark among both organic and inorganic electrode materials. Furthermore, a full-cell configuration using nitro-BTH-COF||NiFeHCF delivers a high-power density and a state-of-the-art energy density of 86.1 Wh kg^{-1} , ranking among the highest reported for NH_4^+ full batteries to date. Overall, the study is comprehensive, well-structured, and supported by extensive experimental and theoretical techniques, including the independent gradient model (IGM) and differential charge density isosurfaces. However, the conclusion section appears somewhat disconnected from the main content of the manuscript and requires a thorough revision. A few other suggestions for improvement are outlined below.

1. The authors have employed Sonogashira coupling to construct the COF. This synthetic strategy is infrequently used in COF development, although it has been previously applied in the synthesis of covalent triazine-based frameworks (CTFs). Could the authors elaborate on the rationale behind choosing this triple-bond-forming linkage over more commonly reported $\text{C}=\text{C}$ bond-forming strategies in COF chemistry? A discussion on the advantages or intended design benefits of this approach would strengthen the justification for its selection.

Response: Thank you for your insightful comments. We have incorporated a discussion regarding the advantages of triple-bond-forming linkage over more commonly reported $\text{C}=\text{C}$ bond-forming strategies in the revised Manuscript: Compared with $\text{C}=\text{C}$ bond-forming strategies that easily induce the spatial structure distortion in COFs, *sp*-conjugated alkyne ($-\text{C}\equiv\text{C}-$) linkages via Sonogashira coupling enables the formation of rigid and robust skeletons to minimize structural deformation of nitro-BTH-COF, which is expected to establish structural anti-dissolution in aqueous electrolytes for durable electrochemical reactions.

2. It would be helpful to include a comparison of the PXRD pattern with those of the starting monomers in the Supporting Information to illustrate the framework formation better.

Response: Thank you for your constructive suggestion. We have included PXRD pattern of the starting monomers of TEB, BTH, and BTH-NO₂ in Supplementary Fig. 6a, and added related discussion behind Supplementary Fig. 6 in revised Supplementary Information: XRD patterns of TEB, BTH, and BTH-NO₂ monomers show diffraction peaks at 10–30°, which completely disappear after Sonogashira coupling reaction, triggering the emergence of distinctive diffraction peaks at 3.5° for highly crystalline BTH-COF and nitro-BTH-COF.

Supplementary Fig. 6 (a) Experimental PXRD patterns of monomers, BTH-COF and nitro-BTH-COF.

3. (Line 139) Please revise the FT-IR peak assignments or elaborate. For example, assigning the peak at 1570 cm⁻¹, as C≡C stretching typically appears around 2100–2200 cm⁻¹. The 1570–1600 cm⁻¹ region is more commonly associated with C=C stretching vibrations, particularly in aromatic systems.

Response: Thank you for your insightful comment. According to a recently reported literature (*J. Am. Chem. Soc.* **2025**, 147, 19667; Ref.52), C≡C stretching typically appears around 2150 cm⁻¹. The 1571 cm⁻¹ region is associated with C=C stretching vibrations in aromatic systems. Thus, we have revised the FT-IR peak assignments in the revised Manuscript: The characteristic peaks at 2155, 1654, 1571 and 1269/1473 cm⁻¹ can be attributed to C≡C bonds, C=N groups, C=C stretching vibrations, and nitro motifs, respectively^{41,52,56-58}.

Fig.1 (f) FT-IR spectra.

4. (Line 29): "Imine bonds" might be more relevant, as they better represent dynamic covalent chemistry than "amine bonds".

Response: Thank you for your valuable suggestion. We have revised the "amine bonds" into "imine bonds" in the revised Manuscript.

5. Please revise the conclusion, as it seems irrelevant to the current manuscript.

Response: Thank you for your valuable suggestion. We have provided a thorough revision of the conclusion to ensure consistency with the data and discussions presented earlier in the revised Manuscript: In conclusion, multiple two-electron nitro-BTH-COF is designed by fusing alkynyl benzenes and nitrobenzothiadiazole units. nitro-BTH-COF delivers highly π -electron *sp*-conjugation along alkynyl linkages and strong electron-drawing nitrobenzothiadiazole motifs. This feature promises high NH_4^+ utilization (95.2%) of multi-two-electron nitro/thiazole sites with a lower activation energy (25.93 vs. 35.99 kJ mol^{-1} of BTH-COF). Experiment and theoretical studies reveal the fast and stable octadeca-H-bonded NH_4^+ coordination mechanism of nitro-BTH-COF anode. As a consequence, nitro-BTH-COF anode liberates the highest capacity among reported COFs in AIBs. Furthermore, the alkynyl-bridged π -conjugation network of nitro-BTH-COF anode gives excellent structural insolubility in aqueous electrolyte, affording unprecedented cycling life. By pairing nitro-BTH-COF anode with high-voltage Prussian blue cathode, the constructed nitro-BTH-COF||NiFeHCF full battery shows state-of-the-art energy density and cycling life. This work gives new insights into the design of multi-electron-redox and stable COFs for advanced AIBs.

6. The equations show that energy density (E) is calculated using the discharge time, current, and the mass of the active substance on a single electrode, rather than the total mass of both electrodes in the full cell. As a result, the reported 86.1 Wh kg^{-1} likely represents an anode-based specific

energy, not a true full-cell value. The authors should clarify the mass basis used for these calculations and, if applicable, recalculate the energy density considering the total active material of both electrodes for a fair comparison with other full-cell systems.

Response: Thank you for your insightful comment. It should be pointed out that the reported energy density (86.1 Wh kg^{-1}) of our full battery is calculated based on the total mass loading of active materials in both anode and cathode. Thus, this result represents the true full-cell specific energy, rather than the anode-based value. We have clarified this point in the revised Manuscript: **Based on the total mass loadings of active materials in nitro-BTH-COF anode (2.2 mg cm^{-2}) and NiFeHCF cathode (5.2 mg cm^{-2}), the high capacity (78 mAh g^{-1}) and high average output voltage (1.1 V) bring a state-of-the-art **battery-level** energy density of 86.1 Wh kg^{-1} among all reported NH_4^+ full batteries (Fig. 5f and Supplementary Table 2)^{43,51,61,71,72}.**

7. Please include the X-ray Photoelectron Spectroscopy (XPS) analysis for the synthesized BTH-COF material. This characterization is essential to confirm the elemental composition and chemical states of the constituent atoms. In particular, XPS spectra should be used to identify bonding interactions and the chemical environment before and after the TEB modification step. Please provide a detailed deconvolution of the relevant peaks to demonstrate the successful modification and to clarify the nature of the new chemical bonds (N-O).

Response: Thank you for your valuable suggestion. We have provided XPS spectra of BTH-COF and nitro-BTH-COF to demonstrate the successful modification and to clarify the nature of the new chemical bonds (N-O) in **Supplementary Fig. 9b** in the revised Supplementary Information.

Supplementary Fig. 9 (b) High-resolution N 1s XPS spectra of nitro-BTH-COF and BTH-COF.

Related discussion was added in the revised Manuscript: **Compared to BTH-COF, the high-resolution N 1s X-ray photoelectron spectrum (XPS) of nitro-BTH-COF exhibits an additional peak**

assigned to NO₂ (Supplementary Fig. 9b), confirming the successful introduction of nitro groups.

8. The authors report high b-values (0.91–0.96, Fig. 2g), indicating a surface-controlled pseudocapacitive process. However, the analysis does not quantitatively separate capacitive and diffusion-controlled contributions. A further capacitive contribution analysis using the relation at various scan rates would provide a clearer understanding of the charge-storage mechanism and strengthen the electrochemical interpretation.

Response: Thank you for your valuable suggestion. We have provided a quantitative analysis to deconvolute the capacitive and diffusion-controlled contributions using the Dunn's method in Supplementary Fig 14a and b, and added related discussion in the revised Manuscript: The capacitive contribution accounts for 71.23% of total charge storage at 0.1 mV s⁻¹ and increases to 93.31% at 1 mV s⁻¹, highlighting its fast redox kinetics (Supplementary Fig. 14a and b).

Supplementary Fig. 14 Charge storage kinetics of nitro-BTH-COF anode. (a) Capacitive contribution. (b) Ratios of capacitive and diffusion-controlled contributions at various scan rates.

9. The text and labels in Figure 3b are not clearly readable. Please improve the resolution, color, and font size of the figure and all annotations.

Response: Thank you for your valuable suggestion. To ensure clear readability, we improved the resolution, color, and font size of the figure and all annotations of Fig. 3b in the revised Manuscript.

Fig. 3 Charge-storage behavior of nitro-BTH-COF anode. (b) Overview of FT-IR spectra.

According to the comments of the expert reviewers, we have thoroughly revised and improved our manuscript to meet the requirements of the journal. Now we resubmit the revised version for your kind consideration.

We would like to express our sincere thanks to you and the expert reviewers for the insightful and constructive comments which significantly contributed to improving the quality of the manuscript.

Yours sincerely,

Prof. Dr. Mingxian Liu

School of Chemical Science and Engineering

Tongji University